

# Is Shale Gas a Major Driver of Recent Increase in Global Atmospheric Methane?

Robert W. Howarth[1]

[1]Department of Ecology & Evolutionary Biology, Cornell University, Ithaca, NY 14853 USA

*Correspondence to*: Robert W. Howarth (howarth@cornell.edu)

**Abstract.** Methane has been rising rapidly in the atmosphere over the past decade, contributing to global climate change. Unlike the late 20th Century when the rise in atmospheric methane was accompanied by an enrichment in the heavier carbon stable isotope ($^{13}$C) of methane, methane in recent years has become more depleted in $^{13}$C. This depletion has been widely interpreted to indicate a primarily biogenic source for the increased methane. Here we show that the change may instead be associated with emissions from shale gas and shale oil development. While methane in conventional natural gas is enriched in $^{13}$C relative to the atmospheric mean, shale gas is depleted in $^{13}$C relative to this atmospheric level. Correcting for this difference, we conclude that emissions from shale gas production in North America over the past decade may well be the leading cause of the increased flux of methane to the atmosphere. Increased fluxes from biogenic sources such as animal agriculture and wetlands are far less important than indicated by some other recent papers using $^{13}$C data.

## 1. Introduction

Methane is the second most important greenhouse gas behind carbon dioxide causing global climate change, contributing approximately 1 watt m$^{-2}$ to warming when indirect effects are included, compared to 1.66 watt m$^{-2}$ for carbon dioxide (IPCC 2013). Unlike carbon dioxide, the climate system responds quickly to changes in methane emissions, and reducing methane emissions could provide an opportunity to immediately slow the rate of global warming (Shindell et al. 2012) and perhaps meet the UNFCC COP21 target of keeping the planet well below 2$^{\circ}$ C above the pre-industrial baseline (IPCC 2018). Methane also contributes to the formation of ground-level ozone, with large adverse consequences for human health and agriculture. Considering these effects as well as climate change, Shindell (2015) estimated the social cost of methane is 40 to 100 times greater than that for carbon dioxide: $2,700 per ton for methane compared to $27 per ton for carbon dioxide when calculated with a 5% discount rate, and $6,000 per ton for methane compared to $150 per ton for carbon dioxide when calculated with a 1.4% discount rate.

Atmospheric methane levels rose steadily during the last few decades of the 20th Century before leveling off for the first decade of the 21st Century. Since 2008, however, methane concentrations have again been rising rapidly (Fig. 1-A). The total atmospheric flux of methane for the period 2008-2014 was ~24.7 Tg per year greater than for the 2000-2007 period



(Worden et al. 2017), an increase of 7% in global human-caused methane emissions. The change in the stable carbon $\delta^{13}C$ ratio of methane in the atmosphere over the past 35 years is striking and seems clearly related to the change in the methane concentration (Fig. 1-B). For the final 20 years of the 20th Century as atmospheric methane concentrations rose, the isotopic composition became more enriched in the heavier stable isotope of carbon, $^{13}C$, relative to the lighter and more abundant

isotope $^{12}C$, resulting in a less negative $\delta^{13}C$ signal. The isotopic composition remained constant from 1998 to 2008 when the atmospheric concentration was constant. And the isotopic composition has become lighter (depleted in $^{13}C$, more negative $\delta^{13}C$) since 2009 as atmospheric methane concentrations have been rising again (Schaefer et al. 2016). Since biogenic sources of methane tend to be lighter than the methane released from fossil-fuel emissions, Schaefer et al. (2016) concluded that the increase in atmospheric methane in the late 20th Century was due to increasing emissions from fossil fuels, but that the

increase in methane since 2006 is due to biogenic sources, most likely tropical wetlands, rice culture, or animal agriculture. Their model results indicated that fossil fuel sources have remained flat or decreased globally since 2006, playing no major role in the recent atmospheric rise of methane. Schaefer et al. (2016) noted that their conclusion contradicts many reports of increased emissions from fossil fuel sources over this time, and stated that their conclusion "is unexpected, given the recent boom in unconventional gas production and reported resurgence in coal mining and the Asian economy." Six months after

the Schaefer et al. (2016) study was published in *Science*, Schwietzke et al. (2016) presented a similar analysis in *Nature* that used a larger and more comprehensive data set for the $\delta^{13}C$ values of methane emissions sources. They too concluded that fossil fuel emissions have likely decreased during this century, and that biogenic emissions are the probable cause of any recent increase in global methane emissions.

## 2. Sensitivity of emission models based on $\delta^{13}C$ in methane to biomass burning

Model analyses that use $\delta^{13}C$ methane data to infer emission sources are highly sensitive to changes in the rate of biomass burning: although biomass burning is a relatively small contributor to global methane emissions, those emissions are quite enriched in $^{13}C$ relative to the atmospheric methane signal (Rice et al. 2016). Both Schaefer et al. (2016) and Schwietzke et al. (2016) assumed that biomass burning had been constant in recent years. However, Worden et al. (2017) estimated that

reductions in biomass burning globally for the period 2007-2014 compared to 2001-2006 resulted in decreased methane emissions of 3.7 Tg per year (± 1.4 Tg per year). Using the data set of Schwietzke et al (2016) for $\delta^{13}C$ values of methane emission sources, but including changes in biomass burning over time, Worden et al. (2017) concluded that the recent increase in methane emissions is likely driven more by fossil fuels than by biogenic sources, with an increase of 15.5 Tg per year from fossil fuels (± 3.5 Tg per year) compared to an increase of 12 Tg per year from biogenic sources (± 2.5 Tg per year) when

comparing 2007-2014 vs 2001-2006.

Clearly global models for partioning methane sources based on the $\delta^{13}C$ approach are sensitive to assumptions about seemingly small terms such as decreases in biomass burning. In this paper, we explore for the first time another assumption:





that the global increase in shale gas development may have caused some of the depletion of $^{13}$C in the global average methane observed over the past decade. Shale gas emissions were not explicitly considered in the models presented by Schaefer et al. (2016) and Worden et al. (2017) and were explicitly excluded in the analysis of Schwietzke et al. (2016).

**3. What is shale gas?**

Shale gas is a form of unconventional natural gas (mostly methane) held tightly in shale rock formations. Conventional natural gas, the dominant form of natural gas produced during the 20$^{th}$ Century, is composed largely of methane that migrated upward from the underlying sources such as shale rock over geological time, becoming trapped under a
geological seal (Fig. 2-A). Until this century, shale gas was not commercially developable. The use of a new combination of technologies in the 21$^{st}$ century – high precision directional drilling, high-volume hydraulic fracturing, and clustered multi-well drilling pads -- has changed this. In recent years, global shale gas production has exploded 14-fold, from 31 billion m$^3$ per year in 2005 to 435 billion m$^3$ per year in 2015 (Fig. 2-B), with 89% of this production in the United States and 10% in Canada (EIA 2016). Shale gas accounted for 63% of the total increase in natural gas production globally over the past decade
(EIA 2016, IEA 2017). The US Department of Energy predicts rapid further growth in shale gas production globally, reaching 1,500 billion m$^3$ per year by 2040 (EIA 2016; Fig. 2-B).

Several studies have shown that the δ$^{13}$C signal of methane from shale gas is often lighter (more depleted in $^{13}$C) than that from conventional natural gas (Golding et al. 2013; Botner et al. 2018). Here, we use the data from Figure 1 in the review
by Golding et al. (2013) that were explicitly identified as shale gas. The samples are from the New Albany shale (Martini et al. 1998), the Antrim shale (McIntosh et al. 2002), and an organic-rich shale in the northern Appalachian basin (Osborn and McIntosh 2010). Note that these studies appear to be the only ones included in the δ$^{13}$C methane data repository published by Sherwood et al. (2017), which is the data set underlying the analysis by Schwietzke et al.(2016). Out of 61 data points for shale gas in the Golding et al. (2013) figure, only 5 had δ$^{13}$C values similar to those for conventional natural gas, while many
samples more closely resembled the signal for biogenic gas. From the 61 values, we calculate a mean value δ$^{13}$C for shale gas of -51.4 $^o$/$_{oo}$ , with a 95% confidence limit of ± 1.2 $^o$/$_{oo}$. Thus, emissions of methane from shale gas are on average depleted in $^{13}$C relative to atmospheric methane, while methane from conventional natural gas is more $^{13}$C-enriched than atmospheric methane.

It should perhaps not be surprising that the δ $^{13}$C of methane from shale gas tends to be lighter than for conventional natural gas. In the case of conventional gas, the methane has migrated over geological time frames from the shale and other source rocks through permeable rocks until trapped below a seal (Fig. 2-A). During this migration, some of the methane is likely oxidized by bacteria, perhaps using iron (III) or sulfate as the source of the oxidizing power (Whelan et al. 1986; Rooze et al. 2016). Partial consumption of methane by bacteria would fractionate the methane by preferentially consuming the lighter



[12]C isotope and so, gradually enriching the remaining methane in [13]C (Baldassare et al. 2014), resulting in a $\delta^{13}$C signal that is less negative. The methane in shales, on the other hand, is tightly held in the rock formation and therefore less likely to have been subject to bacterial oxidation and the resulting fractionation. The expectation, therefore, is that methane in conventional natural gas should be heavier and less depleted in [13]C than is the methane in shale gas.

## 4. Calculating the effect of [13]C signal of shale gas on emission sources

To explore the contribution of methane emissions from shale gas, we build on the analysis of Worden et al. (2017). Figure 3-A shows the $\delta^{13}$C values used by them as well as their mean estimates for changes in emissions since 2008 (as they estimated using the $\delta^{13}$C data of Schwietzke et al. 2016). Figure 3-A represents a weighting for the change in emissions (y-axis) and the $\delta^{13}$C values of those emissions (x-axis) by individual sources. Our addition is to separately consider shale gas emissions, recognizing that methane emissions from shale gas are more depleted in [13]C than for conventional natural gas or all other fossil fuels as considered by Worden et al. (2017). For this analysis, we accept that net total emissions increased by 24.7 Tg per year (± 14. Tg per year) since 2008, driven by an increase of ~28.4 Tg per year for the sum of biogenic emissions and emissions from fossil fuels and a decrease of ~3.7 Tg per year for emissions from biomass burning (Worden et al. 2017).

We start with the Eq. (1) which reweights the information in Figure 3-A for the difference between most fossil fuels and shale gas:

$$12 * D_{B-A} = ( B * D_{B-A} ) + (SG * D_{SG-CG} ) - ( CG * D_{A-CG} ) \tag{1}$$

where 12 Tg per year is the mean estimate from Worden et al. (2017) for the increase in biogenic emissions, $D_{B-A}$ is the difference in the $\delta^{13}$C value for biogenic emission sources and atmospheric methane in 2005, B is the estimate for the increase in biogenic emissions, SG is the estimate for the increase in methane emissions from shale gas, $D_{SG-CG}$ is the difference in the $\delta^{13}$C value for shale gas and conventional natural gas, and DA-CG is the difference in the $\delta^{13}$C value for atmospheric methane in 2005 and for emissions from conventional natural gas. The x-axis of Figure 3-B shows the $\delta^{13}$C for each source; note that the y-axis is the estimate of the change in emissions for each of these sources that we derive below.

As noted above, shale gas accounted for 63% of the global increase in all natural gas production between 2005 and 2015 (EIA 2016, IEA 2017). If we make the simplifying assumption that for both shale gas and conventional natural gas, emissions are equal as a percentage of the gas produced, then

$$SG = 0.63 * TG \tag{2}$$





and

$$CG = 0.37 * TG \qquad (3)$$

where TG is total increase in emissions from all natural gas. Note that we test this assumption later in our sensitivity analyses, since some research indicates emissions from shale gas are higher than for conventional gas as a percentage of gas production. Rearranging Eq. (2) for TG and substituting into Eq. (3),

$$CG = 0.37 * (SG / 0.63), \quad or \qquad CG = 0.59 * SG \qquad (4)$$

Substituting CG from Eq. (4) into Eq. (1),

$$12 * D_{B\text{-}A} = (B * D_{B\text{-}A}) + (SG * D_{SG\text{-}CG}) - (0.59 * SG * D_{A\text{-}CG}) \qquad (5)$$

Next, we estimate the likely contributions from coal and oil to the increased methane emissions over the past decade. We estimate the increase in methane emissions from coal between 2006 and 2016 as 1.3 Tg per year, based on the rise in global coal production of 27%, with almost all of this due to surface-mined coal in China (IEA 2008, 2017), and using a well-accepted emission factor of 870 g methane per ton of surface-mined coal (Howarth et al. 2011). Methane emissions from surface-mined coal tend to be low, as whatever methane was once associated with the coal has degassed over geological time.
This estimate is very close to the 1.1 Tg per year increase from coal emissions in China between 2009 and 2015 as measured from satellite data (Miller et al. 2019). For oil, global production increased by 9.6% (IEA 2008, 2017), thereby increasing methane emissions by approximately 1.6 Tg per year (using emission factors from NETL 2008, as detailed in Howarth et al. 2011). Therefore, of the increase in 28.4 Tg per year from fossil fuels plus biogenic sources since 2005 (see discussion above), we estimate 2.9 Tg per year from increased emissions from coal and oil, leaving an increase of approximately 25.5 Tg per year
from natural gas (including shale gas) plus biogenic sources. That is,

$$SG + CG + B = 25.5 \qquad (6)$$

Substituting CG from Eq. (4) into Eq. (6) and rearranging yields


$$B = 25.5 - (1.59 * SG) \qquad (7)$$

Substituting B from Eq. (7) into Eq. (5),



$$12 * D_{B-A} = \{[25.5 - ( 1.59 * SG )] * D_{B-A} \} + (SG * D_{SG-CG}) - ( 0.59 * SG * D_{A-CG}) \qquad (8)$$

And rearranging to solve for SG,

$$SG = ( 13.5 * D_{B-A} ) / [ ( 1.59 * D_{B-A} ) - ( D_{SG-CG} ) + 0.59 * D_{A-CG}) ] \qquad (9)$$

If we use mean values for the differences in the $\delta^{13}C$ terms in Eq. (9) (ie, $D_{B-A} = 15.35 \permil$, $D_{SG-CG} = 7.4 \permil$, and $D_{A-CG} = 3.15 \permil$), then we estimate the increase in methane emissions from shale gas between 2005 and 2015 (SG) as 11 Tg per year (Table 1). From Eq. (4), increased emissions from conventional natural gas are then estimated as 6.5 Tg per year, and from all natural gas (shale plus conventional) as 17.5 Tg per year. From Eq. (7), increased emissions from biogenic sources are estimated as 8.0 Tg per year. The confidence bounds on these estimates, calculated using Eq. (9) and the upper and lower 95% confidence limits for the $\delta^{13}C$ ratio terms (Fig. 3-B), are relatively small (Table 1).

## 5. Comparison with prior estimates

Our best estimate for the increase in methane emissions from all fossil fuels since 2008 (shale gas, conventional natural gas, coal, and oil) is 20.4 Tg per year (Table 1), substantially larger than the mean estimate of Worden et al. (2017) of 15.5 Tg per year. Our estimate for the increased emissions from biogenic sources, 8 Tg per year, is substantially lower than the Worden et al. (2017) estimate of 12 Tg per year (Table 1). On the other hand, comparing emissions for the 2003-2013 period with those from the late 20[th] Century, Schwietzke et al. (2016) concluded that biogenic emissions had risen by ~ 27 Tg per year while fossil fuel emissions had decreased by ~18 Tg per year. And Schaefer et al. (2016) concluded that increased methane emissions since 2006 are "predominantly biogenic" and that fossil fuel emissions likely have fallen.

Our estimate of increased emissions of 11 Tg per year from shale gas development is quite reasonable in light of the growing body of evidence from measurements made at local to regional scales. Between 2005 and 2015, global shale gas production rose by 404 billion $m^3$ per year (Fig. 2-B) (EIA 2016). Assuming that 93% of natural gas is composed of methane (Schneising et al. 2014), our estimate of the increase in methane emissions from shale gas (11 Tg per year) represents 4.1 % of the increased gas production (an increase of 270 Tg per year of methane produced from shale-gas operations). This estimate of 4.1% (based on global change in the $^{13}C$ content of methane) represents full life-cycle emissions, including those from the gas well site, transportation, processing, storage systems, and final distribution to customers. Our estimate is well within the range reported in several recent studies for shale gas, and in fact is at the low end for many (but not all) of these studies (Howarth et al. 2011; Pétron et al. 2014; Karion et al. 2013; Caulton et al. 2014; Schneising et al. 2014; Howarth 2014). Alvarez et al. (2018) recently presented a summary estimate for natural gas emissions in the United States (both conventional and shale gas) of 2.3% using bottom-up, facility-based data. However, they noted that top-down estimates from approaches





such as airplane flyovers give higher values than the bottom-up estimates they emphasized. In fact, a careful comparison of bottom-up and top-down approaches for one shale-gas field showed 45% higher emissions from the top-down approach, due to under sampling of some emission events by the bottom-up, facility-based approach (Vaughn et al. 2018). Further, Alvarez et al. (2018) used a very low value for the methane emissions from local distribution pipelines, only 0.08 %. Many studies

suggest distribution emissions in Boston, Los Angeles, Indianapolis, and Texas cities may be as high as 2.5% or more, not 0.08 % (Howarth et al. 2011; McKain et al. 2015; Lamb et al. 2016; Wunch et al. 2016), so that a full life-cycle of 4.1% emissions from shale gas over the past decade is quite plausible.

## 6. Sensitivity analyses

Our analysis contain two major assumptions: 1) that methane emissions as a percentage of gas produced are the same for shale gas and conventional natural gas (Eq. (2) and Eq. (3)); and 2) that emissions from oil have remained proportional to the global rate of oil production. Here we explore the sensitivity of our analysis to these assumptions. With regard to the first assumption, some evidence suggests that percent emissions may be higher from shale gas than from conventional natural gas,

perhaps due to venting at the time of flow-back following high-volume hydraulic fracturing of shale-gas wells (Howarth et al. 2011) and also due to release of methane from trapped pockets when drilling down through a very long legacy (often a century or more) of prior fossil fuel operations to reach the deeper shale formations (Caulton et al. 2014; Howarth 2014). For this first sensitivity analysis, we modify equations Eq. (2) through Eq. (9) with new equations Eq. (A1) through Eq. (A7) to reflect a 50% higher emission factor for shale gas than for conventional gas, as proposed in Howarth et al. (2011) (see appendix A).

With this change in assumptions, estimated shale gas emissions increase by 25% (13.9 instead of 11 Tg per year). Biogenic emissions decrease by 23% (6.2 instead of 8 Tg per year), while total fossil fuel emissions increase (22.2 instead of 20.4 Tg per year). The fossil fuel emission estimate is now 3.6-fold larger than the biogenic emission estimate (Table 2).

Our second major assumption in the base analysis is that methane emission factors for oil production have remained

constant over time as a function of production. This may not be true, since 60% of the increase in global oil production between 2005 and 2015 was due to tight oil production from shales using the same technologies that allowed shale gas development, high-precision directional drilling and high-volume hydraulic fracturing (calculated from data in EIA 2015 and EIA 2018). Large quantities of methane are often co-produced with this tight shale oil, and because oil is a much more valuable product than natural gas, for shale-oil fields removed from easy access to natural gas markets, much of the methane may be vented or

flared rather than delivered to market. This may be part of the reason for the large increase in methane emissions in recent years in the Bakken shale fields of North Dakota (Schneising et al. 2014).

For this sensitivity scenario #2, we modify equations Eq. (1) through Eq. (9) with new equations Eq. (B1) through Eq. (B8) to allow for higher emissions associated with shale oil than from conventional oil production (see appendix B). For





this, we follow the approach of Schneising et al. (2014) in combining shale gas and shale oil, scaling the increase in production since 2005 by the energy value of the two products. As in our baseline analysis developed in equations Eq. (1) through Eq. (9), we assume that conventional natural gas and shale gas have the same percentage methane emission per unit of produced gas. Here we further assume that shale oil has the same emission rate as well, scaled to the energy content of oil compared to

natural gas. This sensitivity analysis increases total emissions from fossil fuels by 18% (22.8 instead of 20.4 Tg per year), while biogenic emissions fall (5.6 instead of 8.0 Tg per year) (Table 2). The contribution from shale gas falls somewhat (from 11 to 9.9 Tg per year), as does that from conventional natural gas (from 6.5 to 5.4 Tg per year), while shale oil becomes an important emission source (5.5 Tg per year). Overall in this scenario, increased emissions from fossil fuels extracted from shales (gas plus oil) are 15.4 Tg per year, two-thirds of the total increase due to fossil fuels.

## 7. Conclusions

We conclude that increased emissions from fossil fuels are far more likely than biogenic emissions to have driven the observed global increase in methane over the past decade (since 2008). The increase in emissions from shale gas (perhaps in

combination with those from shale oil) makes up more than half of the total increased fossil fuel emissions. That is, the commercialization of shale gas and oil in the 21st Century has dramatically increased global methane emissions. However, we note an important caveat: our analysis of emissions with explicit consideration of the $\delta\,^{13}C$ value for methane in shale gas is based on a small data set, only 61 samples in 3 studies. A clear priority should be to gather more data on the $^{13}C$ content of shale-gas methane.

Note that while methane emissions are often referred to as "leaks," emissions include purposeful venting, including the release of gas during the flowback period immediately following hydraulic fracturing, the rapid release of gas from blowdowns during emergencies but also for routine maintenance on pipelines and compressor stations (Fig. 4-A), and the steadier but more subtle release of gas from storage tanks (Fig. 4-B) and compressor stations to safely maintain pressures

(Howarth et al. 2011). This suggests large opportunities for reducing emissions, but at what cost? Do large capital investments for rebuilding natural gas infrastructure make economic sense, or would it be better to move to phase natural gas out as fuel and instead invest in a 21st Century energy infrastructure that embraces renewable energy and much more efficient heat and transportation through electrification (Jacobson et al. 2013)?

In October 2018, the Intergovernmental Panel on Climate Change issued a special report, responding to the call of the United Nations COP21 negotiations to keep the planet well below 2º C from the preindustrial baseline (IPCC 2018). They noted the need to reduce both carbon dioxide and methane emissions, and they recognized that the climate system responds more quickly to methane: reducing methane emissions offers one of the best routes to immediately slowing the rate of global warming (Shindell et al. 2012). Nonetheless, the model scenarios presented in the IPCC report emphasize reducing carbon



dioxide emissions first, and these scenarios begin to reduce methane emissions only after 2030. This may reflect the belief of the IPCC authors that methane emissions are dominated by biogenic sources, which are difficult to reduce. Given our conclusion that the oil and gas industry is more likely responsible for recent increases in these emissions, we suggest that the best strategy is to move as quickly as possible away from natural gas, reducing both carbon dioxide and methane emissions.

Doing so will in fact make it easier to reach the COP21 target than predicted by the IPCC (2018).

Finally, in addition to contributing to climate change, methane emissions lead to increased ground-level ozone levels, with significant damage to public health and agriculture. Based on the social cost of methane emissions of $2,700 to $6,000 per ton (Shindell 2015), our baseline estimate for increased emissions from shale gas of 11 Tg per year has resulted in damage

to public health, agriculture, and the climate of $30 to $65 billion USD per year for each of the past several years. This exceeds the wholesale value for this shale gas over these years.

**Appendix A.** Sensitivity case #1: emissions per unit of gas produced assumed to be 50% greater for shale gas than for

conventional gas.

First we modify Eq. (2) and Eq. (3) as follows to reflect that methane emissions per unit of gas produced are 50% greater for shale gas than for conventional natural gas:

$\quad\quad$ SG = 1.2 * (0.63 * TG),   or     $\quad\quad$ SG = 0.76 * TG     $\quad\quad\quad\quad\quad\quad\quad\quad\quad\quad$ (A1)

and

$\quad\quad$ CG = 0.8 * (0.37 * TG),   or     $\quad\quad$ CG = 0.30 * TG     $\quad\quad\quad\quad\quad\quad\quad\quad\quad\quad$ (A2)

Rearranging Eq. (A1) for TG and substituting into Eq. (A2),


$\quad\quad$ CG = 0.30 * ( SG / 0.76),  or     $\quad\quad$ CG = 0.40 * SG     $\quad\quad\quad\quad\quad\quad\quad\quad\quad\quad$ (A3)

Substituting CG from Eq. (A3) into Eq. (1),

$\quad\quad$ $12 * D_{B\text{-}A} = ( B * D_{B\text{-}A} ) + ( SG * D_{SG\text{-}CG} ) - ( 0.40 * SG * D_{A\text{-}CG} )$     $\quad\quad\quad\quad\quad\quad$ (A4)

Substituting CG from Eq. (A3) into Eq. (6),

$\quad\quad$ SG + ( 0.40 * SG) + B = 25.5,      or with rearrangement,



$$B = 25.5 - ( 1.40 * SG ) \tag{A5}$$

Substituting B from Eq. (A5) into Eq. (A4),

$$12 * D_{B-A} = \{[25.5 - ( 1.40 * SG )] * D_{B-A} \} + (SG * D_{SG-CG} ) - ( 0.30 * SG * D_{A-CG} ) \tag{A6}$$

And rearranging to solve for SG,

$$SG = ( 13.5 * D_{B-A} ) / [ ( 1.40 * D_{B-A} ) - ( D_{SG-CG} ) + 0.30 * D_{A-CG} ) ] \tag{A7}$$

If we use mean values for the differences in the $\delta^{13}C$ terms in Eq. (A7) (as we did previously for Eq. (9)), for shale gas, SG = 13.8 Tg per year. From Eq. (A3), for conventional natural gas, CG = 5.5 Tg per year. From Eq. (A5), for biogenic emissions, B = 6.2 Tg per year. Total fossil fuel emissions are estimated as the contributions from coal (1.3 Tg per year) and

15 oil (1.6 Tg per year) plus SG plus CG, or 22.2 Tg per year. These values are reported in Table 2.

**Appendix B**. Sensitivity case #2: explicit consideration of shale oil (tight oil).

For the base analysis presented in the main text using equations Eq. (1) through Eq. (9), we assumed that increased emissions from the additional oil development over the past decade were proportional to the increase in that rate of development. That is, the oil produced in recent years had the same emission factor as for oil produced a decade or more ago. However, 60% of the increase in oil production globally between 2005 and 2015 was for tight oil from shale formations

25 (calculated from data in EIA 2015 and EIA 2018), and methane emissions from this shale oil may be greater than for conventional oil. In this sensitivity case #2, we consider increased emissions from conventional oil and from tight shale oil separately. For conventional oil, the increase in emissions is 40% of the total oil emissions from the base analysis (40% of 1.6 Tg per year, or 0.65 Tg per year, rounded to 0.7 in Table 2), reflecting that conventional oil contributed 40% to the growth in oil production between 2005 and 2015.

30

For the tight shale oil, we follow the approach used by Schneising et al. (2014): the increase in methane emissions from shale gas and shale oil are considered together, normalized to the energy content of the two fuels. Shale gas production increased by 405 billion $m^3$ per year between 2005 and 2015 (EIA 2016). With an energy content of 37 MJ per $m^3$, this reflects an increase in 15.9 trillion MJ per year. For shale oil, production increased by 230 liters per year between 2005 and 2015





(EIA 2015, 2018). With an energy content of 38 MJ per liter, this reflects an increase in 8.9 trillion MJ per year. Conventional natural gas production increased by 238 billion m$^3$ per year between 2005 and 2015 (EIA 2016). With an energy content of 37 MJ per m$^3$, this reflects an increase in 8.8 trillion MJ per year. Therefore, the sum of the increase in production for shale gas, shale oil, and conventional natural gas is 33.6 trillion MJ per year. Shale gas represents 48% of this, shale oil 26%, and

5  conventional natural gas represents 26%. The sum of shale gas and shale oil represents 74% of the total.

For this sensitivity analysis, we further assume that shale gas and conventional natural gas have the same percentage emissions, as in our base case analysis in the main text, and that the $^{13}$C content of methane from shale oil is the same as for shale gas. Using these assumptions, we modify Eq. (2) and Eq. (3) as follows,

$$\text{SG\&O} = 0.74 * \text{TG\&SO} \tag{B1}$$

and

$$\text{CG} = 0.26 * \text{TG\&SO} \tag{B2}$$

where SG&O is shale gas plus shale oil and TG&SO is total natural gas plus shale oil. Rearranging Eq. (B1) for TG&SO and substituting into Eq. (B2),

$$\text{CG} = 0.26 * (\text{SG\&O} / 0.74), \text{ or} \qquad \text{CG} = 0.35 * \text{SG\&O} \tag{B3}$$

Substituting CG from Eq. (B3) into Eq. (1) modified by substituting SG&O for SG,

$$12 * D_{B\text{-}A} = (B * D_{B\text{-}A}) + (\text{SG\&O} * D_{\text{SG-CG}}) - (0.35 * \text{SG\&O} * D_{A\text{-CG}}) \tag{B4}$$

25

Modifying Eq. (6) for SG&O rather than SG, and estimating 1.95 Tg per year from increased emissions from coal and conventional oil, leaves an increase of approximately 26.45 Tg per year from natural gas (including shale gas) plus tight shale oil plus biogenic sources, so

30

$$\text{SG\&O} + \text{CG} + \text{B} = 26.45 \tag{B5}$$

Substituting CG from Eq. (B3) into Eq. (N5),

$$\text{SG \&O} + (0.35 * \text{SG\&O}) + \text{B} = 26.45, \qquad \text{or with rearrangement,}$$



$$B = 26.45 - ( 1.35 * SG\&O ) \tag{B6}$$

Substituting B from Eq. (B6) into Eq. (B4),

$$12*D_{B-A} = \{[26.45 - 1.35* SG\&O )]*D_{B-A} \}+(SG\&O *D_{SG-CG} )-( 0.35 * SG\&O * D_{A-CG} ) \tag{B7}$$

Rearranging to solve for SG&O,

10 $$SG\&O = ( 14.45 * D_{B-A} ) / [ ( 1.35 * D_{B-A} ) - ( D_{SG-CG} ) + 0.35 * D_{A-CG} ) ] \tag{B8}$$

If we use mean values for the differences in the $\delta^{13}C$ terms in Eq. (B8) (as we did previously for Eq. (9) and Eq. (A7)), for shale gas plus shale oil emissions, SG&O = 15.4 Tg per year. From Eq. (B3), for conventional natural gas, CG = 5.4 Tg per year. From Eq. (B6), for biogenic emissions, B = 5.6 Tg per year.

Total fossil fuel emissions are estimated as the contributions from coal (1.3 Tg per year), conventional oil (0.65 Tg per year), conventional natural gas (5.4 Tg per year), plus the sum for shale gas plus shale oil (15.4 Tg per year), or 22.8 Tg per year. We can separately estimate shale gas and shale oil, estimating the proportion of the sum of the two made up by shale gas as follows,

$$15,9 \text{ trillion MJ per yr} / (15.9 \text{ trillion MJ per year} + 8.9 \text{ MJ per year}) = 0.64 \tag{B9}$$

Therefore, of the 15.4 Tg per year for SG&O, shale gas emissions are 9.9 Tg per year and shale oil emissions are 5.5 Tg per year. These values are reported in Table 2.

25

30

## 8. Author contributions and competing interests



Robert Howarth is the sole author, responsible for all aspects of this work. The author declares that he has no conflicts of interest.

## 9. Acknowledgements

Financial support was provided by the Park Foundation and an endowment given by David R. Atkinson to support the professorship at Cornell University held by RWH. We thank Tony Ingraffea, Amy Townsend-Small, Euan Nisbet, Martin Manning, Dennis Swaney, and Roxanne Marino for comments on earlier versions of this manuscript. We particularly thank Dennis Swaney for helpful discussion and review of the analyses we report. We thank Gretchen Halpert for the art work in Figures 1 and 2; Sharon Wilson for the photographs in Figure 4-A; and Jack Ossont for the photograph in Figure 4-B. Tony Ingraffea helped interpret these photographs.

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



Table 1. Estimates for sources of increased or decreased methane emissions to the atmosphere in recent years (Tg per year). All values are positive, except as specified.

| | This study [a] | Worden et al. [b] (2017) | Schwietzke et al. [c] (2016) |
|---|---|---|---|
| **All fossil fuels** | **20.4** | **15.5** (± 3.5) | **negative ~ 18** |
| -- shale gas | 11.0 (± 0.7) | | |
| -- conventional gas | 6.5 (± 0.4) | | |
| -- coal | 1.3 | | |
| -- oil | 1.6 | | |
| **Biogenic sources** | **8.0** (± 1.0) | **12.0** (± 2.5) | **~ 27** |

[a] Time period is 2008-2014 compared to 2000-2007. Confidence bounds for estimates for shale gas, conventional natural gas, and biogenic sources are calculated using Eq. (9) and the upper and lower 95% confidence limits for the $\delta^{13}C$ values shown in Figure 3-B.

[b] Time period is 2008-2014 compared to 2000-2007, with values from their Figure 4, with the Schwietzke et al. (2016) data set and assuming a decrease in biomass burning of 3.7 Tg per year. Uncertainty is as shown in original publication.

[c] Time period is for 2003-2013 compared to 1985-2002, with values from their Figure 2.B. Uncertainties are large, and only mean differences shown here.




Table 2. Exploration of sensitivity to assumptions for estimates of increase in global methane emissions in recent years (Tg per year). Only mean values presented.

| | Base analysis [a] | Increased emission factor for shale gas (sensitivity test #1) [b] | Explicit consideration of shale oil (sensitivity test #2) [c] |
|---|---|---|---|
| **All fossil fuels** | **20.4** | **22.2** | **22.8** |
| -- all natural gas | 17.5 | 19.3 | 15.3 |
| -- shale gas | 11.0 | 13.8 | 9.9 |
| -- conventional gas | 6.5 | 5.5 | 5.4 |
| -- all oil | 1.6 | 1.6 | 6.2 |
| -- shale oil | | | 5.5 |
| -- conventional oil | | | 0.7 |
| -- coal | 1.3 | 1.3 | 1.3 |
| **Biogenic sources** | **8.0** | **6.2** | **5.6** |

[a] Base analysis is from equations Eq. (1) through Eq. (9) and is also presented in Table 1. Assumptions include equivalent percentage emissions as a function of production for shale gas and conventional natural gas, and no contribution of $^{13}$C-depleted methane from tight shale oil production.

[b] Same assumptions as for the base analysis, except shale gas emissions are assumed to be 50% greater than those from conventional natural gas, expressed as a percentage of production.

[c] Same assumptions as for the base analysis, except emission of $^{13}$C-depleted methane from shale oil is explicitly considered.



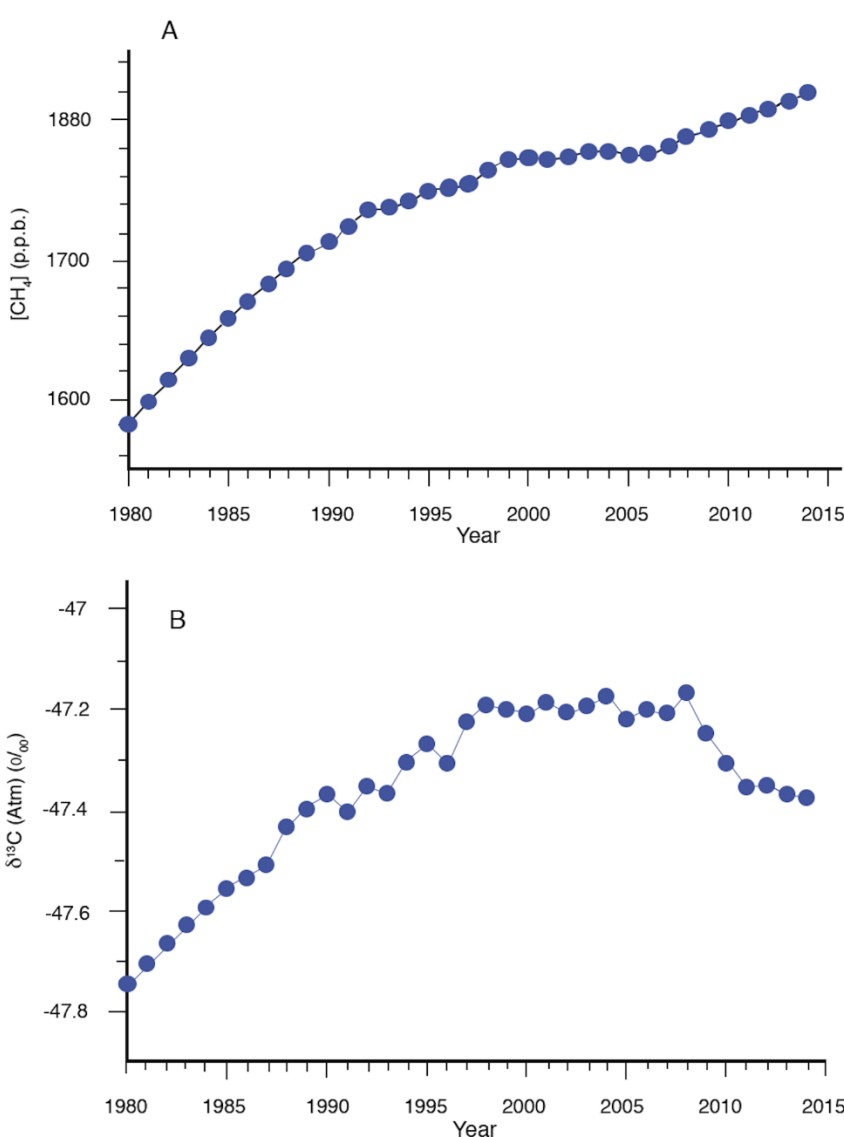

Figure 1. A, top: global increase in atmospheric methane between 1980 and 2015. B, bottom: change in $\delta^{13}$C value of atmospheric methane globally between 1980 and 2015. Both adapted from Schaefer et al. (2015).



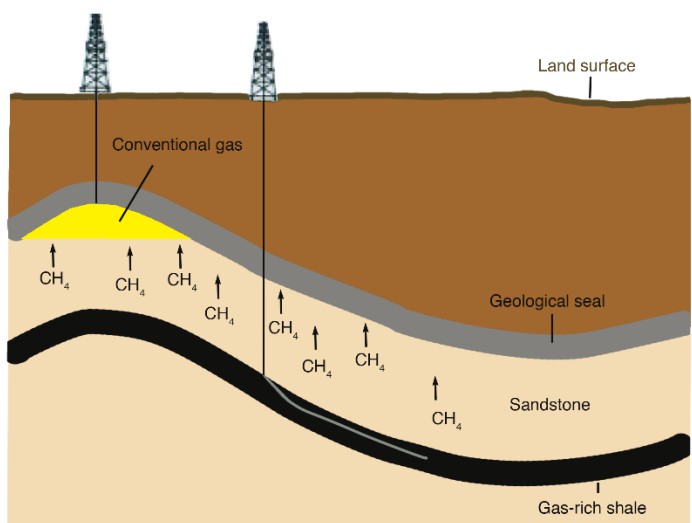

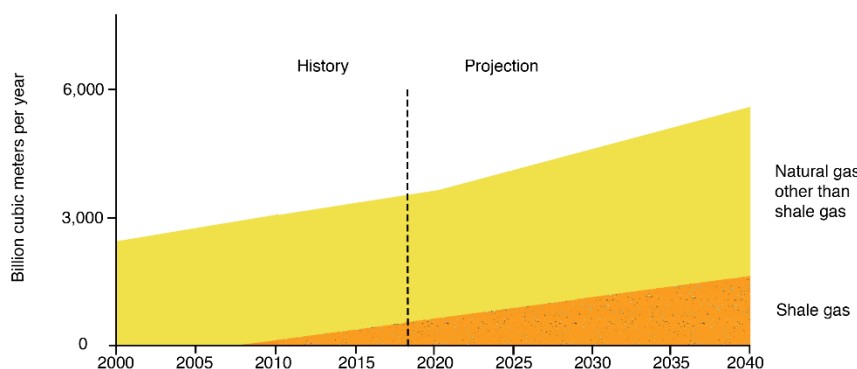

Figure 2. A, top: schematic comparing shale gas and conventional natural gas. For conventional natural gas, methane migrates

from the shale through semi-permeable formations over geological time, becoming trapped under a geological seal. Shale gas

5   is methane that remained in the shale formation and is released through the combined technologies of high-precision directional

drilling and high-volume hydraulic fracturing.

B, bottom: global production of shale gas and other forms of natural gas from 2000 through 2017, with projections into the

future from EIA (2016). Redrawn from EIA (2016) with data from IEA (2017).





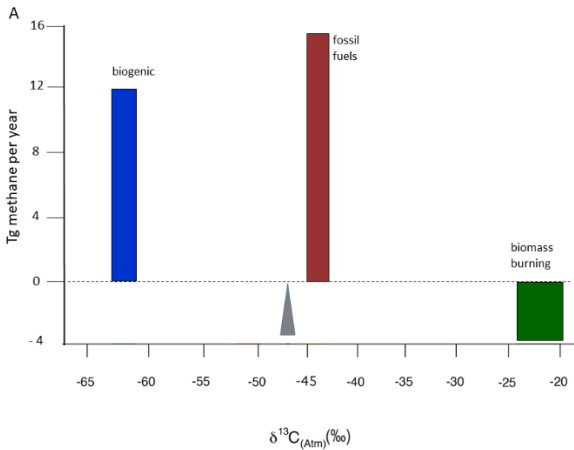

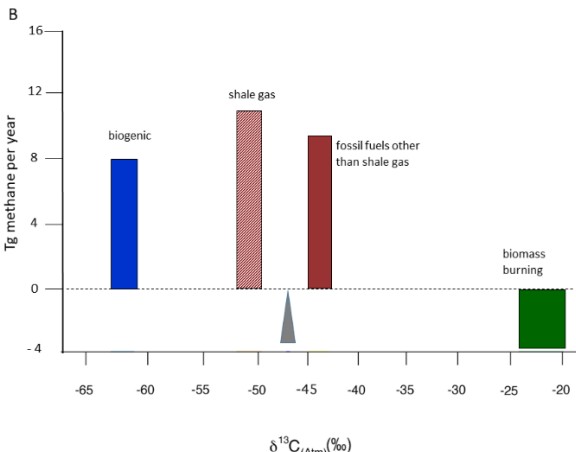

5  Figure 3. A, top: on the x-axis,$\delta^{13}$C values for methane from biogenic sources, fossil fuel, and biomass burning as presented in Worden et al. (2017) for values from Schwietzke et al. (2016); width of horizontal bars represent the 95% confidence limits for these values. Triangle indicates the average methane in the atmosphere during the 2000 to 2008 period. The y-axis shows mean estimates from Worden et al. (2017) for the increase and decrease in methane emissions from particular sources since 2008 as calculated using the $\delta^{13}$C values of Schwietzke et al. (2016).

10  B, bottom: on the x-axis, $\delta^{13}$C values as in Figure 3-A except the value for fossil fuels does not include shale gas, and a separate estimate for shale gas value is included based on the mean and 95% confidence level for 61 observations from three different shales and studies summarized by Golding et al. (2013): mean value is -51.4 ‰ , with a 95% confidence limit of ± 1.2 ‰. The y-axis indicates estimates developed in this paper for the increase or decrease in methane emissions since 2008.





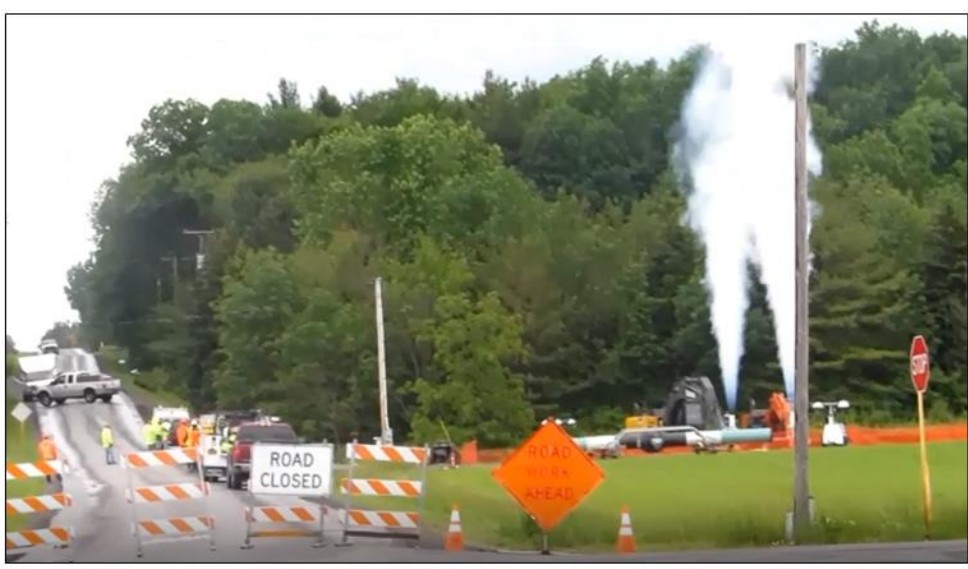

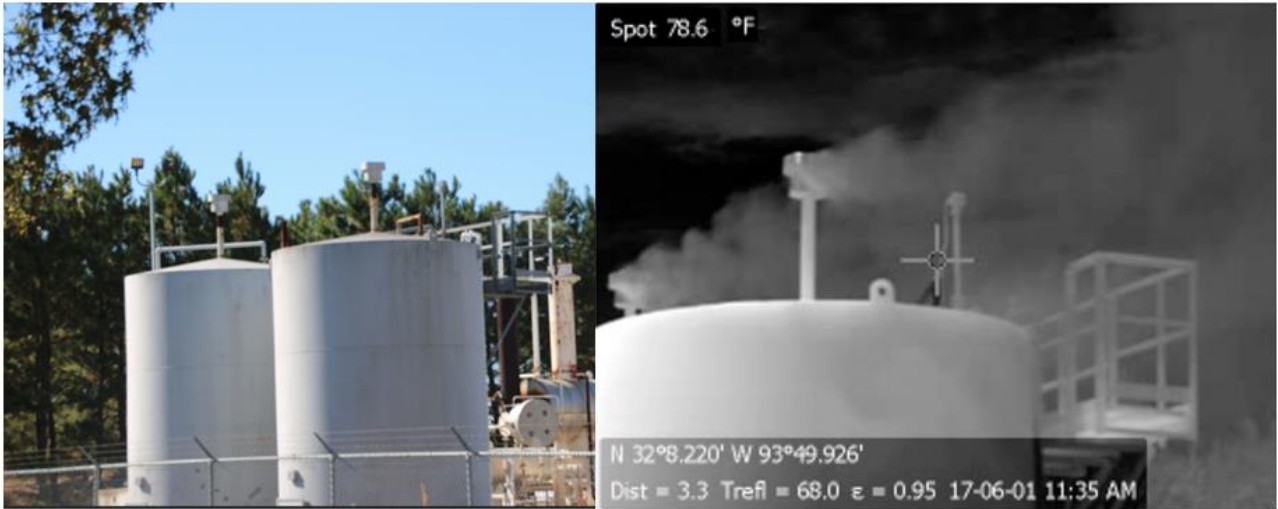

Figure 4. A, top: gas blowdown for maintenance on a pipeline in Yates County, NY. While methane is invisible, the cooling

5    caused by the blowdown condenses water vapor, leading to the obvious cloud. Photo courtesy of Jack Ossont.

B, bottom: Gas storage tanks receiving natural gas from feeder pipelines before compression for transport in high-pressure

pipelines, Haynseville shale formation, Texas. Photo on left was taken with a normal camera. Photo on the right was taken

with a FLIR camera tuned to the infra-red spectrum of methane, allowing visualization of methane, which is invisible in the

normal camera view and to the naked eye. Photo courtesy of Sharon Wilson.