# Peer review of "Is Shale Gas a Major Driver of Recent Increase in Global Atmospheric Methane?"

_Biogeosciences, 2019_

## Referee Comment (RC1) · Anonymous Referee #1 · 3 May 2019

This manuscript takes up the widely discussed question of the origin of the renewed methane growth in recent years by analyzing the atmospheric $\delta^{13}$C methane isotopologue signature. As $\delta^{13}$C has decreased in recent years, several studies attributed the corresponding methane increase to biogenic sources, arguing that biogenic methane is more depleted in $^{13}$C than the average atmospheric value during the hiatus period at the beginning of the century, while methane from fossil fuels is not.

The author tries to revise previous estimates of the shares of biogenic sources and fossil fuels by explicitly distinguishing between shale gas and conventional gas claiming that the isotopologue signature from shale gas resembles biogenic gas more closely than conventional gas. Although the idea sounds promising at first glance, the representativity of the chosen shale gas plays for the entire sector is questionable and

a revised and more comprehensive data set is needed to recommend publication in Biogeosciences and to support the far-reaching conclusions drawn by the author.

**General Comments**

The main weakness of the analysis is that the $\delta^{13}$C-methane value of $(-51.4 \pm 1.2)$‰ for shale gas is derived from a small data set from the review paper Golding et al.(2013) consisting of 61 samples in 3 studies, which is postulated here to be representative for the entire shale gas production sector.

In contrast to that, Tilley et al.(2013) present new and review former isotope data from several published papers including shale gas from the Barnett Shale, Fayetteville Shale, Marcellus Shale, Utica Shale and the Western Canada Sedimentary Basin. All of the various presented samples are less depleted than $-51.4$‰ and almost all are even less depleted than $-47.2$‰ which is the average $\delta^{13}$C-methane during the 2000-2008 period according to Figure 1B. Hence, these samples cannot explain the depletion of $\delta^{13}$C in the atmosphere since 2009 and do not point to a shale gas origin.

Furthermore, Tilley et al. identify three common maturation stages of shale gas systems and point out that the so called *rollover zone* may represent the peak of high productivity shale gas. The rollover zone roughly corresponds to $\delta^{13}$C-methane values between $-45$‰ and $-35$‰ for the analyzed cases. This would support the implicit assumption of Worden et al.(2017) that the isotopologue signature of shale and conventional gas is similar.

**Specific Comments**

Page 1, Lines 11-12: The statement that shale gas is depleted in $^{13}$C relative to the atmospheric mean is not supported by sufficient evidence (see also general comments).

Page 3, Lines 18-19: There are also several studies (Tilley et al., 2013 and references therein) which do not support this statement. Moreover, the cited paper of Bottner et al. assumes a value of $-47.3$‰ for shale gas, which is larger than the $-51.4$‰ value used

for the presented analysis and comparable to the average atmospheric $\delta^{13}$C-methane during the 2000-2008 period.

Page 6, Lines 16-19: Apart from the question of representativity of the presented analysis, is the derived difference to the Worden et al.(2017) estimates really significant? I find it hard to believe that the uncertainties of this study presented in Table 1 are that small although it is ultimately a reweight of the Worden et al. estimates. Are the prior uncertainties from Worden et al. considered correctly?

Page 7, Lines 30-31: This sentence creates the impresssion that the methane emissions in the Bakken shale are steadily increasing. However, it seems that after years of considerable increase (Schneising et al., 2014), emissions have been reduced again (Peischl et al., 2018).

Conclusions: The advice to move as quickly as possible away from natural gas based on this study does not appear sufficiently conclusive for the reasons mentioned above. A thorough analysis of the impact of shale gas and the adequacy of natural gas as a bridge fuel is highly desirable, but to draw such strong conclusions based on a small data set, which likely lacks representativity, is premature.

**Technical Corrections**

Several instances: there is no space ahead of $\delta^{13}$C. Please check.

Page 4, Line 25: DA-CG should be D$_{A-CG}$

Page 22, Line 7: Triangle indicates the average $\delta^{13}$C-methane...

**References**

Tilley, B. and Muehlenbachs, K.: Isotope reversals and universal stages and trends of gas maturation in sealed, self-contained petroleum systems, Chemical Geology, 339, 194-204, https://doi.org/10.1016/j.chemgeo.2012.08.002, 2013.

Peischl, J., Eilerman, S. J., Neuman, J. A., Aikin, K. C., de Gouw, J., Gilman, J. B.,

et al.: Quantifying methane and ethane emissions to the atmosphere from central and western U.S. oil and natural gas production regions. Journal of Geophysical Research: Atmospheres, 123, 7725–7740, https://doi.org/10.1029/2018JD028622, 2018.

———————————————————

---

## Author Comment (AC1) · 9 May 2019

Author response to anonymous review #1 (as of 9 May 2019):

I thank the reviewer for their comments and suggestions, which I will carefully consider as I revise my manuscript. Below are my preliminary responses.

Major Criticism – choice of value for $\delta$13C content of methane from shale gas: The greatest criticism of the reviewer is my reliance on the studies reviewed by Golding et al. (2013) for the estimate of the $\delta$13C content of methane from shale gas. My preliminary responses to this are:

1) the reviewer is correct that the 13C value used for the shale gas methane is critical to my analysis. However, I believe I have fairly and accurately reported representative

data in my discussion draft, albeit from an admittedly small set of studies. Nonetheless, as I revise the manuscript, I will consider further data as appropriate;

2) note that I submitted my manuscript under the "Ideas & Perspectives" category and not as a regular research paper. My intent was to draw attention to the failure of earlier papers that evaluated changes in atmospheric methane using the $\delta$13C approach to explicitly consider that shale gas may differ from conventional natural gas, but I recognized the limited data behind my re-evaluation;

3) in that context, one of the major conclusions of my manuscript (lines 18-21, p. 8) was that "... we note an important caveat: our analysis of emissions with explicit consideration of the $\delta$13C value for methane in shale gas is based on a small data set, only 61 samples in 3 studies. A clear priority should be to gather more data on the 13C content of shale-gas methane."

4) for the $\delta$13C data I used, I chose the studies reviewed in Golding et al. (2013) because these were highlighted by Schwietzke et al. (2016) (one of the key earlier papers on changes in the global $\delta$13C value of methane) as representative for shale gas. Note that Schwietzke et al. (2016) did not include these shale gas data (either explicitly as distinct from conventional natural gas or as part of their overall estimate for fossil fuels), stating in their supplemental materials: "Note that $\delta$13CFF in this analysis excludes shale gas methane because the share of these sources to global total NG production increased from only 3% to 9% between 2007 and 201316." Reference #16 is to Golding et al. (2013). As an aside, I believe the important context for looking at changes in methane over the past decade is not the increase from 3% to 9% of total natural gas production, but rather the fact that "shale gas accounted for 63% of the global increase in all natural gas production between 2005 and 2015 (EIA 2016, IEA 2017)" as stated in my manuscript (lines 29-30, p. 4).

5) I welcome new data on the $\delta$13C value for shale gas, and I thank the reviewer for pointing out the Tilley & Muehlenbachs paper. However, the Tilley & Muehlenbach

paper appears to report data both for actual shale gas (that is, the methane trapped in shale which is released by high-volume hydraulic fracturing) and for gas trapped in reservoirs that has migrated from the shale over geological time scales (which would be considered "conventional" gas in my manuscript). Before I revise my manuscript, I will consult with experts to ascertain which of the data in the Tilley & Muehlenbach paper might be relevant for my analysis (that is, those representing actual shale gas that would be developed from high-volume hydraulic fracturing).

Major Criticism – Tilley & Muehlenbach support no difference between shale and conventional gas: In the final paragraph of their General Comments, the reviewer states "… Tilley et al. identify three common maturation stages of shale gas systems and point out that the so called rollover zone may represent the peak of high productivity shale gas. The rollover zone roughly corresponds to $\delta$ 13C-methane values between $-45‰$ and $-35‰$ for the analyzed cases. This would support the implicit assumption of Worden et al. (2017) that the isotopologue signature of shale and conventional gas is similar." I disagree that the language in Tilley & Muehlenbach supports the implicit assumption of Worden et al. (2017) that shale gas and conventional gas are indistinct in their 13C content. Rather, I believe the preponderance of evidence indicates that shale gas is more depleted in 13C than is conventional gas. Specifically:

1) I find the reviewer's argument speculative, and note that Tilley & Muehlenbach (2013) state "….. the mechanism that created the reversed gases is still not well understood and is controversial."

2) as noted above, Tilley & Muehlenbach are reporting data both for actual shale gas (as defined in my paper) and for conventional gas that has migrated from shale gas (which I would expect to be more enriched in 13C; see point #4 below);

3) in contrast to the argument made by the reviewer, some of the sites with the most negative $\delta$13C-methane values presented in the Tilley & Muehlenbach paper occur in the Marcellus shale; the Marcellus shale has been the largest shale-gas play in the

world over the past decade and has accounted for more than 25% of all shale gas production globally over this time period (EIA 2018). Further, these 13C-methane data for the Marcellus shale as reported in Tilley & Muehlenbach are apparently for methane that has migrated from the shale to a reservoir. If so, this methane is likely to be more enriched (heavier, less negative $\delta$13C) than the actual shale gas (see point #4 below);

4) as I wrote in my discussion paper (line 30 on p. 3 through line 4 on p. 4, and Fig. 2-A), one should expect that methane that migrates out of shale and is trapped in reservoirs would be more enriched in 13C than the methane that remains in the shale: some of the methane is consumed by bacteria as it migrates, with fractionation by the bacteria causing this enrichment.

Specific Comments

"Page 1, Lines 11-12: The statement that shale gas is depleted in 13C relative to the atmospheric mean is not supported by sufficient evidence (see also general comments)."

Please see my detailed responses to the first "major criticism" above.

"Page 3, Lines 18-19: There are also several studies (Tilley et al., 2013 and references therein) which do not support this statement. Moreover, the cited paper of Bottner et al. assumes a value of $-47.3$‰ for shale gas, which is larger than the $-51.4$‰ value used for the presented analysis and comparable to the average atmospheric $\delta$13C-methane during the 2000-2008 period."

There are two parts to this criticism. With regard to whether or not the Tilley & Muehlenbach paper supports my statement, please again refer to my detailed responses above to major criticism #1. Regarding Botner et al. (2018), that paper clearly states that the methane in shale gas is likely to be more depleted in 13C relative to conventional natural gas, as I stated in my discussion manuscript (line 18 to 19, p. 3). Yes, their shale gas estimate is heavier than the mean I used in my analysis (again, based on the papers reviewed in Golding et al. 2013) but is not statistically outside of the range I used.

Please note that before I submitted this manuscript to Biogeosciences, I consulted with a co-author of the Botner et al. (2018) paper (Amy Townsend-Small), sharing with her a draft of what I submitted.

"Page 6, Lines 16-19: Apart from the question of representativity of the presented analysis, is the derived difference to the Worden et al.(2017) estimates really significant? I find it hard to believe that the uncertainties of this study presented in Table 1 are that small although it is ultimately a reweight of the Worden et al. estimates. Are the prior uncertainties from Worden et al. considered correctly?"

The uncertainty for the Worden et al. estimates presented in my Table 1 come from that paper. For my new estimates in Table 1, the uncertainty is estimated as stated in the footnote: "Confidence bounds for estimates for shale gas, conventional natural gas, and biogenic sources are calculated using Eq. (9) and the upper and lower 95% confidence limits for the $\delta$13C values shown in Figure 3-B." I leave it to the reader to decide whether these new estimates are significantly different from those presented in Worden et al. (2017). There certainly is at least a trend, with my estimates strongly suggesting greater emissions from fossil fuels and smaller biogenic emissions.

One take home point for my discussion manuscript – which remember is in the "Ideas & Perspectives" category and is not a traditional research paper – is that small changes in assumptions can lead to major changes in the conclusions from these global analyses. Worden et al. (2017) showed that decreases in biomass burning over the past decade (Schaefer et al. 2016 and Schwietzke et al. 2016 had assumed a constant rate of biomass burning) qualitatively changed the conclusion on trends in fossil-fuel emissions: rather than fossil fuel emissions going down over the past decade by some 18 Tg/yr (Schwietzke et al. 2016, and my Table 1), they more likely increased by 15.5 Tg/yr (plus or minus 3.5 Tg/yr), according to Worden et al. (2017). My inclusion of a consideration that shale gas methane may be more depleted in 13C than the methane in conventional gas further increases the estimate for the change in emissions from fossil fuels, to a mean estimate of 20.4 Tg/yr greater emissions (my Table 1). There are

large uncertainties associated with any of these estimates, but the directional change in estimates from changes in assumptions is very clear.

"Page 7, Lines 30-31: This sentence creates the impression that the methane emissions in the Bakken shale are steadily increasing. However, it seems that after years of considerable increase (Schneising et al., 2014), emissions have been reduced again (Peischl et al., 2018)."

The time of reference for my analysis, following that of Worden et al. (2017), was 2005 to 2015. The Schneising et al. (2014) reference is quite pertinent to that time frame, as they compared 2006-2008 to 2009-2011. Nonetheless, as I revise, I will note that some evidence suggests emissions in Bakken shale may have decreased over the past few years (Peischl et al. 2018).

"Conclusions: The advice to move as quickly as possible away from natural gas based on this study does not appear sufficiently conclusive for the reasons mentioned above. A thorough analysis of the impact of shale gas and the adequacy of natural gas as a bridge fuel is highly desirable, but to draw such strong conclusions based on a small data set, which likely lacks representativity, is premature."

I appreciate this comment, but again, my contribution is as an ""Ideas & Perspectives" piece. To date, previous analyses that used the change in the $\delta$13C value for global methane over time have completely ignored the possible role of shale gas, even though shale gas was 63% of the total global increase in natural gas over the past decade. I do not view my manuscript as a final statement: it is a call to pay better attention to this potentially critical aspect of the story as to what is driving the increase in global methane emissions. And I note that my conclusion is qualified, as stated above: "an important caveat: our analysis of emissions with explicit consideration of the $\delta$13C value for methane in shale gas is based on a small data set, only 61 samples in studies. A clear priority should be to gather more data on the 13C content of shale-gas methane" (lines 18-21, p. 8).

Both Schaefer et al. (2016) and Schwietzke et al. (2016) in very high profile papers (Science and Nature) concluded that methane emissions from fossil fuel sources had declined over the past decade, even though Schaefer et al. (2016) noted how surprising this conclusion was in the context of the massive global increase in unconventional (shale) oil and gas development. My analysis suggests a critical element these authors may have missed in their analyses. To point out this omission is overdue, in my opinion, not premature.

Technical Corrections I thank the reviewer for these technical corrections.

---

## Referee Comment (RC2) · Anonymous Referee #2 · 10 May 2019

This study represents a novel, valuable contribution to the ongoing discussion of the causes of the renewed growth in observed methane concentrations in the atmosphere. In my opinion it is very appropriate for the Ideas and Perspectives of this journal. I have one major question and one major suggestion, both below, along with several minor comments that I believe would improve the paper.

General comments

1) I have tried at some length, but I cannot understand equation 1. Figure 3A shows the weighting used by Worden et al, whereas Figure 3B has the new weighting used here. My understanding is that equation 1 converts the results of Worden et al shown in Fig 3A into the new division of Fig 3B. It makes sense to neglect biomass burning here as that's assumed to be the same in both. I don't understand, however, why

the Worden et al estimate for biogenic (left side of equation 1) would be equal to the redistributed sum including the total CG term. For example, if there were no shale gas production (SG=0), this equation should maintain the Worden et al results, but it seems to me it doesn't as the CG*DA-CG term would still be there. Should not the 'CG' in that equation actually represent the change in CG in the reweighted values compared with the Worden et al value rather than the entire CG emissions? Then the equation would represent the revised biomass term, the shift due to the additional SG term, and the decreased allocation to CG, which should sum to the total of the original biomass plus CG from Figure 3A.

2) In addition to using isotopic data to identify the source of the recent increase in observed methane concentrations, the other information that previous studies have used is the geographic location of observed trends. This can help determine if the source is likely tropical (and hence probably biogenic) or from Northern Hemisphere mid-latitudes (and hence more plausibly with a substantial fossil share). This paper doesn't address this issue, and while it doesn't contribute new knowledge in this area it would be good for the reader to have a short discussion of results from this line of inquiry and how those compare with the conclusions drawn here. For example, Nisbet et al claim their box model suggests much of the increase is from tropical or Southern latitudes. The Rice et al study (already cited) found conflicting results, however. Similarly, at least some studies using satellite observations have suggested that increases are largely at mid-latitudes (e.g. Schneising et al; Turner et al).

Additional References: Nisbet, E. G., et al. (2016), Rising atmospheric methane: 2007–2014 growth and isotopic shift, Global Biogeochem. Cycles, 30, 1356–1370, doi:10.1002/2016GB005406.

Turner, A. J., et al., Geophys. Res. Lett., 43, 2218, 2016.

Additional comments:

P1, L22: UNFCC should be UNFCCC

P1, L29: This is the first mention of Fig 1A. This has an error in the y-axis labels, which show 1880 where it should be 1800.

P2, L1: There should be a space before the delta symbol, here and hereafter (e.g. P2,L21; P3,L18, etc.).

P3, L25: How are the 61 data points weighted, all the same? It there is uneven sampling, is it necessary to weight by geographic location to avoid bias (e.g. giving equal weight to the three regions mentioned previously)?

P4, L25: In the text reading DA-CG, the 'A-CG' portion should be subscript.

P5, L20-21: Should say something like 'estimated based on satellite observations' rather than 'as measured from satellite data' as the satellite cannot measure any specific source of methane emissions, only total methane concentration.

P8, L34-P9, L1: The text here states that "the model scenarios presented in the IPCC report emphasize reducing carbon dioxide emissions first, and these scenarios begin to reduce methane emissions only after 2030." This is incorrect. The scenarios are designed to achieve long-term targets at least cost, and as methane reductions are often very cost-effective these occur fairly rapidly in most scenarios. For example, Figure SPM.3a shows methane emissions relative to 2010 in the 1.5C scenarios, and the midpoint of the range is about a 40% decrease by 2030. Reductions are indeed typically larger and more rapid for CO2, but methane drops quite substantially early on.

P8, L1-L2: The phrase 'This may reflect the belief of the IPCC authors that methane emissions are dominated by biogenic sources' is not an appropriate way to describe characteristics of the scenarios in the SR1.5. The scenarios do not reflect beliefs of the authors, but rather results from integrated assessment models that the authors analyzed. Language such as "This may reflect an overestimate of the fraction of methane emissions attributed to biogenic sources in the underlying integrated assessment models" would be much better.

P9, L7-11: Some related calculations were shown in Shindell, D., J. S. Fuglestvedt, W. J. Collins, The Social Cost of Methane: Theory and Applications, Faraday Disc., 200, 429-451, doi: 10.1039/C7FD00009J, 2017, which could be noted here.

---

## Referee Comment (RC3) · Anonymous Referee #3 · 10 May 2019

Manuscript Summary This manuscript attempts to re-examine the role of fossil emissions in the recent rise of atmospheric methane. This is an important topic that has been the subject of vigorous scientific debate. The author argues that emissions from shale gas are considerably more depleted in 13C than conventional gas, which should change the estimates of relative contributions from different methane sources in an isotopic mass balance calculation. Explicitly including shale gas in the isotopic analysis and making some assumptions about the shale: conventional gas ratio, the author arrives at revised estimates for contributions from fossil (higher than prior estimates) and biogenic (lower than prior estimates) sources to the recent methane increase.

Overall Evaluation: Explicitly including information about the d13C signature of shale gas into an isotope mass balance calculation to re-assess today's global methane budget seems like a useful contribution, if it has not been done already. However, unfortunately this study contains a major flaw in the isotope mass balance analysis that invalidates its results.

Specific Comments: 1. The approach of comparing methane source isotopic signatures directly to the mean global atmospheric d13CH4 value (graphically illustrated by Fig 3 in the manuscript and used in Equation 1) is conceptually flawed. The mean atmospheric d13CH4 is NOT the same as the emission-weighted sum of source d13C signatures. The reason for this is that there is a relatively large isotopic fractionation associated with the atmospheric sink of CH4, with 12CH4 being removed faster than 13CH4. This results in a $\sim$ 6 to 7 per mil enrichment of mean d13C of atmospheric methane with respect to the mean d13C of global methane emissions. To put it another way, for a steady-state mean atmospheric d13CH4 of -47.2 per mil (value during the pause in atmospheric CH4 rise in early 2000s), the mean d13C of global CH4 emissions is $\sim$ -53.7 per mil. This is an effect that has been known for a very long time and is incorporated into all recent atmospheric methane budget analyses that use d13C, including the Schwietzke et al. (2016, Nature) and Schaefer et al. (2016, Science) papers the author cites. When this atmospheric fractionation effect is taken into account, even the much more negative (compared to conventional gas) d13C value of -51.4 per mil proposed by the author for shale gas is still higher than mean d13C of global emissions. Increases in emissions from shale gas would therefore still drive atmospheric d13CH4 up, not down. The author needs to re-do their analysis using a proper isotopic mass balance approach that incorporates the CH4 sink fractionation and also accounts for the fact that methane in the atmosphere today is not in steady-state. Examples of such calculations are described in detail in the supplements to the Schwietzke and Schaefer papers mentioned above, among others.

2. Equation 1 is a strange approach to isotopic mass balance, and it is difficult to judge whether or not it is correct (the issue above aside). The author should either provide a detailed derivation of their form of this equation to illustrate why it's valid or use a

more conventional isotopic mass balance approach – again see the Schwietzke and Schaefer papers for examples.

3. Page 3, line 25. How representative is this d13C value (which seems to be based on a limited number of measurements and sites) of the cumulative shale gas emissions? Is this a simple arithmetic mean? Is it possible to estimate an emissions-weighted mean (which would be more appropriate for an isotope mass balance calculation)? The 95% confidence limit stated seems very narrow to me.

---

## Author Comment (AC2) · 3 Jun 2019

Author response to anonymous review #2 (3 June 2019):

I greatly appreciate the kind words of the reviewer in finding my study novel and valuable. The reviewer has one major question, one major suggestion, and several more minor additional comments. I respond to each of these below.

GENERAL COMMENT #1: "I have tried at some length, but I cannot understand equation 1. Figure 3A shows the weighting used by Worden et al, whereas Figure 3B has the new weighting used here. My understanding is that equation 1 converts the results of Worden et al shown in Fig 3A into the new division of Fig 3B. It makes sense to neglect biomass burning here as that's assumed to be the same in both. I don't un-

[Figure]

derstand, however, why the Worden et al estimate for biogenic (left side of equation 1) would be equal to the redistributed sum including the total CG term. For example, if there were no shale gas production (SG=0), this equation should maintain the Worden et al results, but it seems to me it doesn't as the CG*DA-CG term would still be there. Should not the 'CG' in that equation actually represent the change in CG in the reweighted values compared with the Worden et al value rather than the entire CG emissions? Then the equation would represent the revised biomass term, the shift due to the additional SG term, and the decreased allocation to CG, which should sum to the total of the original biomass plus CG from Figure 3A."

The reviewer is correct. In the revised manuscript, I have completely rewritten the approach, deriving new equations, and better explaining the logic. The new language follows:

"To explore the contribution of methane emissions from shale gas, we build on the analysis of Worden et al. (2017). Figure 3-A shows the $\delta$13C values used by them as well as their mean estimates for changes in emissions since 2008 (as they estimated using the $\delta$13C data of Schwietzke et al. 2016). Figure 3-A represents a weighting for the change in emissions (y-axis) and the $\delta$13C values of those emissions (x-axis) by individual sources. Our addition is to separately consider shale gas emissions, recognizing that methane emissions from shale gas are more depleted in 13C than for conventional natural gas or all other fossil fuels as considered by Worden et al. (2017). For this analysis, we accept that net total emissions increased by 24.7 Tg per year ($\pm$ 14. Tg per year) since 2007, driven by an increase of $\sim$28.4 Tg per year for the sum of biogenic emissions and emissions from fossil fuels and a decrease of $\sim$3.7 Tg per year for emissions from biomass burning (Worden et al. 2017).

"We start with the Eq. (1) which explicitly considers methane emissions from shale gas:

( BN - BW ) + ( FFN - FFW ) + SG = 0 (1)

where BN is the estimate from Worden et al. (2017) for the increase in biogenic emissions of methane globally after 2007, BW is our new estimate for the increase in these biogenic fluxes, FFN is the estimate from Worden et al. (2017) for the increase in emissions of methane globally from fossil fuels after 2007, FFW is our new estimate for the increase in fossil fuel emissions after 2007 other than from shale gas, and SG is our estimate for emissions from shale gas after 2007. That is, the inclusion of an estimate for shale gas is matched by changes in the estimated fluxes from biogenic sources and other fossil fuels.

"Eq. (2) then reweights the information in Figure 3-A for the difference between most fossil fuels and shale gas, multiplying global mass fluxes for each source by the difference between the $\delta$13C ratio of each source and the flux-weighted mean for all sources:

( BN - BW ) * DB-A = [ ( FFN - FFW ) * DA-FF ) ] + ( SG * DA-SG) (2)

where DB-A , DFF-A, and DSG-A are the differences in the $\delta$13C ratio of biogenic emissions, fossil fuels, and shale gas compared to the flux-weighted mean $\delta$13C ratio for all sources (A). The x-axis of Figure 3-B shows the $\delta$13C for each source; note that the y-axis is the estimate of the change in emissions for each of these sources that we derive below. Next, if we multiply both sides of equation 1 by DB-A and rearrange,

( BN - BW ) * (DB-A ) = - [ ( FFN - FFW ) * ( DB-A ) ] - ( SG * DB-A ) (3)

"Subtracting equation 3 from equation 2,

0 = [ ( FFN - FFW ) * ( DA-FF + DB-A ) ] + [ SG * ( DA-SG + DB-A ) (4)

"Rearranging equation 4,

SG = - ( FFN - FFW ) * ( DA-FF + DB-A ) / ( DA-SG + DB-A ) (5)

"Note that from Worden et al. (2017), FFN is 16.4 Tg per year."

From here, the text closely follows that in the "discussion" manuscript, except using

updated values in response to comments from reviewers #1 and #3.

MAJOR SUGGESTION: "2) In addition to using isotopic data to identify the source of the recent increase in observed methane concentrations, the other information that previous studies have used is the geographic location of observed trends. This can help determine if the source is likely tropical (and hence probably biogenic) or from Northern Hemisphere mid-latitudes (and hence more plausibly with a substantial fossil share). This paper doesn't address this issue, and while it doesn't contribute new knowledge in this area it would be good for the reader to have a short discussion of results from this line of inquiry and how those compare with the conclusions drawn here. For example, Nisbet et al claim their box model suggests much of the increase is from tropical or Southern latitudes. The Rice et al study (already cited) found conflicting results, however. Similarly, at least some studies using satellite observations have suggested that increases are largely at mid-latitudes (e.g. Schneising et al; Turner et al). Additional References: Nisbet, E. G., et al. (2016), Rising atmospheric methane: 2007–2014 growth and isotopic shift, Global Biogeochem. Cycles, 30, 1356–1370, doi:10.1002/2016GB005406. Turner, A. J., et al., Geophys. Res. Lett., 43, 2218, 2016."

This is a good suggestion, and I will bring this information into the revised manuscript as I rewrite, including the two satellite papers (Schneising et al. and Turner et al.), the Nisbet et al. (2016) paper and also a new Nisbet et al. (2019) paper. Regarding Rice et al., please note that their analysis ends in 2009, just at the time the shale gas revolution was starting, and so their findings are not applicable to my paper; I will make mention of this in my revised manuscript.

ADDITIONAL COMMENTS:

"P1, L22: UNFCC should be UNFCCC"

Thanks, correction made.

[Figure]

"P1, L29: This is the first mention of Fig 1A. This has an error in the y-axis labels, which show 1880 where it should be 1800."

Correction made.

"P2, L1: There should be a space before the delta symbol, here and hereafter (e.g. P2,L21; P3,L18, etc.)."

Oddly, the spacing is fine on my Word version, but obviously is wrong on the generated pdf. I will try to fix this.

"P3, L25: How are the 61 data points weighted, all the same? It there is uneven sampling, is it necessary to weight by geographic location to avoid bias (e.g. giving equal weight to the three regions mentioned previously)?"

Yes, all points were weighed equally. Since the different studies all had a similar number of points, this is not a large issue, although I agree with the reviewer it would be better to weight them. However, in response to a comment from reviewer #1, I am no longer relying on these 61 data points. Please see my second response to reviewer #1 (posted 3 June 2019).

"P4, L25: In the text reading DA-CG, the 'A-CG' portion should be subscript."

Correction made.

"P5, L20-21: Should say something like 'estimated based on satellite observations' rather than 'as measured from satellite data' as the satellite cannot measure any specific source of methane emissions, only total methane concentration."

Good point; I will make this revision.

"P8, L34-P9, L1: The text here states that "the model scenarios presented in the IPCC report emphasize reducing carbon dioxide emissions first, and these scenarios begin to reduce methane emissions only after 2030." This is incorrect. The scenarios are designed to achieve long-term targets at least cost, and as methane reductions are

often very cost-effective these occur fairly rapidly in most scenarios. For example, Figure SPM.3a shows methane emissions relative to 2010 in the 1.5C scenarios, and the midpoint of the range is about a 40% decrease by 2030. Reductions are indeed typically larger and more rapid for CO2, but methane drops quite substantially early on."

I have deleted this text.

"P8, L1-L2: The phrase 'This may reflect the belief of the IPCC authors that methane emissions are dominated by biogenic sources' is not an appropriate way to describe characteristics of the scenarios in the SR1.5. The scenarios do not reflect beliefs of the authors, but rather results from integrated assessment models that the authors analyzed. Language such as "This may reflect an overestimate of the fraction of methane emissions attributed to biogenic sources in the underlying integrated assessment models" would be much better."

I have deleted this text.

"P9, L7-11: Some related calculations were shown in Shindell, D., J. S. Fuglestvedt, W.J. Collins, The Social Cost of Methane: Theory and Applications, Faraday Disc., 200, 429-451, doi: 10.1039/C7FD00009J, 2017, which could be noted here."

Thank you for this lead.

---

## Author Comment (AC3) · 3 Jun 2019

UPDATED RESPONSE BY AUTHOR TO REVIEWER #1 (3 June 2019):

Please see my first response (dated 9 May 2019) for specific responses to the reviewer's comments. Since writing that response, I have spent considerable time scouring the literature for additional appropriate data on the 13C content of shale gas, and I have substantially changed the value I use in my analysis (from - 51.4 o/oo to – 46.9 o/oo).

In addition to the original sources I used in the "discussion" submission (which were reviewed in Golding et al. 2013), I followed the leads in the Tilley and Muehlenbachs (2013) review suggested by the reviewer, as well as those in the Sherwood et al. (2017)

[Figure]

data set. With regard to the work cited by Tilley and Muehlenbach (2013), some of these studies refer to methane that has migrated from the original shale formation, and not to methane that would be released from shale through high-volume hydraulic fracturing (which is how I and most others define "shale gas"). Since my argument is that the methane would be subject to fractionation by partial oxidation during migration, it would not be appropriate to include data on these migrated gases. Included in the Tilley and Muehlenbach (2013) paper are data from Tilley et al. (2011): note that Hao and Zou (2013) specifically decided not to include those data in their modeling, noting that fractionation during migration seemed likely. Similarly, many of the samples listed by Sherwood et al. (2017) as "shale" are not in fact not for shale gas that is released through high-volume hydraulic fracturing, but rather again for methane that has migrated from shales. In some cases, it is possible to determine from the original papers cited whether or not the samples are truly for shale gas, but in many cases this is not possible.

My response is to only use data for samples that unambiguously came from shale gases, and that clearly were not from migrated gases. One such set of data come from Botner et al. (2018), which reviewer #1 specifically suggested I consider. Further, I have decided not to use the organic-rich shales from the Golding et al. (2013) review – as I had done in the "discussion" submission – as I now feel these are unlikely to reflect the major shale gas plays of the past decade. In rewriting, I am now including additional papers on 13C fractionation from partial oxidation of migrating methane.

I have extensively revised two paragraphs in my revised submission. The new language follows:

[revised manuscript text omitted]

---

## Author Comment (AC4) · 3 Jun 2019

Author response to anonymous review #3 (3 June 2019):

I thank the reviewer for their comments, and I am glad they agree it is useful to explicitly consider shale gas in using data on the 13C content of methane to evaluate methane emission sources over time. Below I respond to each comment.

"Overall Evaluation: Explicitly including information about the d13C signature of shale gas into an isotope mass balance calculation to re-assess today's global methane budget seems like a useful contribution, if it has not been done already. However, unfortunately this study contains a major flaw in the isotope mass balance analysis that invalidates its results."

[Figure]

No previous work has explicitly considered the 13C signal of shale gas when modeling changes in atmospheric sources of methane. In fact, Schwietzke et al. (2016) specifically excluded shale gas from their study. As for the flaw, I disagree that this is a fatal problem, although I do agree it should be addressed. See further discussion below after "specific comments: 1."

"Specific Comments: 1. The approach of comparing methane source isotopic signatures directly to the mean global atmospheric d13CH4 value (graphically illustrated by Fig 3 in the manuscript and used in Equation 1) is conceptually flawed. The mean atmospheric d13CH4 is NOT the same as the emission-weighted sum of source d13C signatures. The reason for this is that there is a relatively large isotopic fractionation associated with the atmospheric sink of CH4, with 12CH4 being removed faster than 13CH4. This results in a ∼ 6 to 7 per mil enrichment of mean d13C of atmospheric methane with respect to the mean d13C of global methane emissions. To put it another way, for a steady-state mean atmospheric d13CH4 of -47.2 per mil (value during the pause in atmospheric CH4 rise in early 2000s), the mean d13C of global CH4 emissions is ∼ -53.7 per mil. This is an effect that has been known for a very long time and is incorporated into all recent atmospheric methane budget analyses that use d13C, including the Schwietzke et al. (2016, Nature) and Schaefer et al. (2016, Science) papers the author cites. When this atmospheric fractionation effect is taken into account, even the much more negative (compared to conventional gas) d13C value of -51.4 per mil proposed by the author for shale gas is still higher than mean d13C of global emissions. Increases in emissions from shale gas would therefore still drive atmospheric d13CH4 up, not down. The author needs to re-do their analysis using a proper isotopic mass balance approach that incorporates the CH4 sink fractionation and also accounts for the fact that methane in the atmosphere today is not in steady-state. Examples of such calculations are described in detail in the supplements to the Schwietzke and Schaefer papers mentioned above, among others."

I thank the reviewer for their very clear explanation of why I should have used a

weighted mean value for the methane entering the atmosphere, and not the average value for the current atmospheric methane. In the revised manuscript, I have followed their guidance. In the revision, I now have written "The average $\delta$13C ratio for methane in the atmosphere in 2005 was - 47.15 o/oo (Schneising et al. 2016), which reflects a flux-weighted mean input of methane with a $\delta$13C ratio of – 53.5 o/oo. This flux-weighted mean value is approximately 6.3 o/oo more depleted in 13C because of fractionation during the oxidation of methane in the atmosphere (Schneising et al. 2016; Sherwood et al. 2017). In our analysis, we use this flux-weighted mean value of – 53.5 o/oo."

Note that the effect of this change on my calculations and conclusions is not large: the important part of the analysis is that the shale gas is more depleted in 13C than are the values used for fossil fuels by Schaefer et al. (2016), Scwietzke et al. (2016), and Worden et al. (2017).

"2. Equation 1 is a strange approach to isotopic mass balance, and it is difficult to judge whether or not it is correct (the issue above aside). The author should either provide a detailed derivation of their form of this equation to illustrate why it's valid or use a more conventional isotopic mass balance approach – again see the Schwietzke and Schaefer papers for examples."

Reviewer #2 also expressed concern over my equation 1. In response, I have completely modified the approach and supporting language. The new text reads:

"To explore the contribution of methane emissions from shale gas, we build on the analysis of Worden et al. (2017). Figure 3-A shows the $\delta$13C values used by them as well as their mean estimates for changes in emissions since 2008 (as they estimated using the $\delta$13C data of Schwietzke et al. 2016). Figure 3-A represents a weighting for the change in emissions (y-axis) and the $\delta$13C values of those emissions (x-axis) by individual sources. Our addition is to separately consider shale gas emissions, recognizing that methane emissions from shale gas are more depleted in 13C than for

conventional natural gas or all other fossil fuels as considered by Worden et al. (2017). For this analysis, we accept that net total emissions increased by 24.7 Tg per year ($\pm$ 14. Tg per year) since 2007, driven by an increase of $\sim$28.4 Tg per year for the sum of biogenic emissions and emissions from fossil fuels and a decrease of $\sim$3.7 Tg per year for emissions from biomass burning (Worden et al. 2017).

"We start with the Eq. (1) which explicitly considers methane emissions from shale gas:

( BN - BW ) + ( FFN - FFW ) + SG = 0 (1)

where BN is the estimate from Worden et al. (2017) for the increase in biogenic emissions of methane globally after 2007, BW is our new estimate for the increase in these biogenic fluxes, FFN is the estimate from Worden et al. (2017) for the increase in emissions of methane globally from fossil fuels after 2007, FFW is our new estimate for the increase in fossil fuel emissions after 2007 other than from shale gas, and SG is our estimate for emissions from shale gas after 2007. That is, the inclusion of an estimate for shale gas is matched by changes in the estimated fluxes from biogenic sources and other fossil fuels.

"Eq. (2) then reweights the information in Figure 3-A for the difference between most fossil fuels and shale gas, multiplying global mass fluxes for each source by the difference between the $\delta$13C ratio of each source and the flux-weighted mean for all sources:

( BN - BW ) * DB-A = [ ( FFN - FFW ) * DA-FF ) ] + ( SG * DA-SG ) (2)

where DB-A , DFF-A, and DSG-A are the differences in the $\delta$13C ratio of biogenic emissions, fossil fuels, and shale gas compared to the flux-weighted mean $\delta$13C ratio for all sources (A). The x-axis of Figure 3-B shows the $\delta$13C for each source; note that the y-axis is the estimate of the change in emissions for each of these sources that we derive below. Next, if we multiply both sides of equation 1 by DB-A and rearrange,

( BN - BW ) * (DB-A ) = - [ ( FFN - FFW ) * ( DB-A ) ] - ( SG * DB-A ) (3)

"Subtracting equation 3 from equation 2,

0 = [ ( FFN - FFW ) * ( DA-FF + DB-A ) ] + [ SG * ( DA-SG + DB-A ) (4)

"Rearranging equation 4,

SG = - ( FFN - FFW ) * ( DA-FF + DB-A ) / ( DA-SG + DB-A ) (5)

"Note that from Worden et al. (2017), FFN is 16.4 Tg per year."

From here, the text closely follows that in the "discussion" manuscript, except using updated values in response to a comment from reviewer #1, and the change the use of the weighted mean value for methane entering the atmosphere (discussed under specific comment #1, above).

"3. Page 3, line 25. How representative is this d13C value (which seems to be based on a limited number of measurements and sites) of the cumulative shale gas emissions? Is this a simple arithmetic mean? Is it possible to estimate an emissions-weighted mean (which would be more appropriate for an isotope mass balance calculation)? The 95% confidence limit stated seems very narrow to me."

I have revised my choice of the d13C value for methane in response to reviewer #1, now using a value of – 46.9 o/oo, which is a weighted mean for three values from major shale-gas plays: the Bakken (North Dakota), Barnett (Texas), and Utica (Ohio). The value is more enriched than the - 51.4 o/oo value chose for my original submission; I now feel that value is too negative, and the organic-rich shales which were included there are probably not representative of most shale gas that has been developed over the past decade. Please see my detailed response to reviewer #1 (updated, 3 June 2019) for my logic.

---

## Author Comment (AC5) · 2 Jul 2019

Department of Ecology and Evolutionary Biology

July 1, 2019

Prof. Jack Middleburg
Associate Editor
*Biogeosciences*

Dear Jack:

I have extensively revised my submission #bg-2019-131 entitled "Is shale gas a major driver of recent increase in global atmospheric methane" in response to the comments of three reviewers. I found the comments of all of the reviewers to be constructive, and I feel my submission is substantially improved. I have previously written responses to each of the reviews (two responses, in the case of reviewer #1), and these have been posted on the web page for my discussion article since the 3rd of June. Below, I indicate in detail how I have revised my manuscript in light of these reviewer comments, but I do not repeat the detailed responses I provided earlier. I am submitting two versions of the revision, one showing full edit mode with all of the changes clearly indicated, and another where all of the changes have been accepted, making it easier to read the final version. The take-home message of my paper remains largely unchanged from that of my original submission.

I look forward to hearing your response.

Best regards,

Robert W. Howarth, Ph.D.
*David R. Atkinson Professor of Ecology
and Environmental Biology*
* * *
**List of revisions made:**

Page 1, abstract: In response to reviewer #1, I am now using a different value for the d13C of methane for shale gas; this is not as different from the value for conventional natural gas, and I have changed the language here to reflect that. I have also somewhat toned down the conclusions.

Page 2, lines 4-6: Reviewer #2 had suggested I include mention to Nisbet et al. (2016); I have done so, but also found a more recent relevent paper, Nisbet et al. (2019); that paper makes an important point about the urgency of controlling methane emissions, and I have added reference to it here.

Page 2, line 14: I have added the Nisbet et al. (2016) reference here.

Page 2, line 29: I have added Sherwood et al. (2017) as a reference here, to strengthen the importance of getting biomass burning right in the 13C analysis.

Page 3, line 26, through page 14, line 14: In response to the comment by reviewer #1, I have changed the d13C value used for shale gas, which has resulted in extensive changes in the language here (as well as later, see new paragraph on page 6, lines 2-19). Note that I have added several new references that support that methane is partially oxidized and therefore fractionated as it migrates from shales into conventional reservoirs.

Page 4, line 28, through page 5, line 32; page 6, line 27 through page 7, line 15; and page 7, line 32, through page 9, line 5: In response to comments by both reviewer #2 and reviewer #3, I have substantially altered the equations used to estimate shale gas emissions.

Page 6, line 2-19: as noted above, I have added a new paragraph here, presenting a new value for the d13C of shale gas. I believe my logic is fully explained in this paragraph, as well as in the updated June 3 reply to reviewer #1.

Pge 6, lines 21-25: reviewer #3 argued that my analysis should be based not on the d13C value for methane currently in the atmosphere, but rather on the mass-weighted mean value for the methane emitted to the atmosphere that would lead to that value, after fractionation due to the oxidation sink in the atmosphere. I have taken this advice, and presented the new approach in this paragraph, with supporting references.

Page 6, line 30: I added a new header, to break up this rather long section.

Page 7, line 23: as suggested by reviewer #2, I have changed "measured" to "estimated base don."

Page 9, lines 9-17: I have made some minor adjustments in language to reflect the change in results from the the new conceptual approach and the new d13C for methane from shale gas.

Page 9, lines 19-34: at the suggestion of reviewer #2, I have added a new paragraph to discuss how my results inform the discussion over the changes in methane emission geographically since 2007. As I discuss in my response to reviewer #2, I do not include reference here to Rice et al., as he had suggested, as their analysis ends in 2009, just as the shale gas revolution was accelerating.

Page 10, lines 2-21: I have made a few small changes here, mostly reflecting the new results. I deleted language on line 17 in response to a comment I received orally during the question section of a talk I gave on this paper in Amsterdam on June 13 at the 8th International Symposium on Non-CO2 Greenhouse Gases.

Page 11, line 1-6: I have made small changes in response to the new results.

Page 11, line 15: I have modified this language in response to the comment by the reviewer suggesting that emissions from the Bakken shale have decreased in the past few years. The Schneising et al. study only evaluates emissions through 2011.

Page 11, lines 17-28: I have made small changes in response to the new results.

Page 12, lines 2-4: I deleted some text to reflect the use of a new value for the mean d13C content of shale gas.

Page 12, lines 19-25: I have deleted some text in response to comments by reviewer #2.

Pge 12, ines 29-31:  I have made some small changes to reflect the new results.

Page 13, line 15, through page 14, line 9:  these calculations are changed to reflect the new conceptual approach used.

Page 15, line 16, through page 16, line 27:  these calculations are change to reflect the new conceptual approach used.

References, page 17 through page 22:  I added 14 new references to support the choice of d13C for shale gas methane and in response to suggestions made by the reviewers.

Tables 1 and 2, pages 24 and 25:  numbers are updated to reflect the new results.

Figure 1-A, page 26:  1880 was changed to 1800 on the y-axis, correcting an error caught by reviewer #2.

Figures 3-A and 3-B, pages 30 and 31:  these figures are updated to reflect the new results and to use the mean mass-weighted d13C for all emission sources, as suggested by reviewer #3.

Note that other figures remain unchanged, but I needed to re-enter them into the draft due to some issue with Microsoft Word.

[revised manuscript text omitted]
 after 2007.  That is, the inclusion of an estimate for shale gas is matched by changes in the estimated fluxes from biogenic sources and other fossil fuels.

5             (1)

10

Eq. (2) then reweights the information in Figure 3-A for the difference between most fossil fuels and shale gas, multiplying global mass fluxes for each source by the difference between the $\delta^{13}C$ ratio of each source and the flux-weighted
15 mean for all sources:

$$( B_N - B_W )  *  D_{B-A}  =  [ ( FF_N  -  FF_W )  *  D_{A-FF} ) ]  +  ( SG  *  D_{A-SG} )  \qquad (2)$$

where $D_{B-A}$, $D_{FF-A}$, and $D_{SG-A}$ are the differences in the $\delta^{13}C$ ratio of biogenic emissions, fossil fuels, and shale gas compared
20 to the flux-weighted mean $\delta^{13}C$ ratio for all sources (A).  The x-axis of Figure 3-B shows the $\delta^{13}C$ for each source; note that the y-axis is the estimate of the change in emissions for each of these sources that we derive below.  Next, if we multiply both sides of equation 1 by $D_{B-A}$ and rearrange,

$$( B_N - B_W ) * (D_{B-A} ) = - [ ( FF_N  -  FF_W ) * ( D_{B-A} ) ]  -  ( SG  * D_{B-A} )  \qquad (3)$$

Subtracting equation 3 from equation 2,

$$0 = [ ( FF_N  -  FF_W )  *  ( D_{A-FF}  +  D_{B-A} ) ]  +  [ SG  * ( D_{A-SG}  + D_{B-A} )  \qquad (4)$$

30 Rearranging equation 4,

$$SG = -  ( FF_N  -  FF_W ) * ( D_{A-FF}  +  D_{B-A} )  / ( D_{A-SG}  + D_{B-A} )  \qquad (5)$$

Note that from Worden et al. (2017), $FF_N$ is 16.4 Tg per year.

Although our expectation is that the methane in shale gas is depleted in $^{13}$C relative to conventional natural gas, the $\delta^{13}$C ratios for the methane in both conventional gas reservoirs and in shale gas vary substantially, changing with the maturity of the gas and several other factors (Golding et al. 2013; Hao and Zou 2013; Tilley and Muehlenbachs 2013). The large data

5  set of Sherwood et al. (2017) suggests no systematic difference between the average ratio for shale gas and the average for conventional gas. However, some of the data listed as shale gas in that data set are actually for methane that has migrated from shale to reservoirs (Tilley et al. 2011) and therefore may have been partially oxidized and fractionated (Hao and Zou 2013). In other cases, the data appear to come both from conventional vertical wells and shale-gas horizontal wells in the same region, making interpretation ambiguous (Rodriguez and Philp 2010; Zumberge et al. 2012). Note that in the Barnett shale

10  region, Texas, the $\delta^{13}$C ratio for methane emitted to the atmosphere ( - 46.5 $^o\!/_{oo}$; Townsend-Small et al. 2015) is more depleted than the average for wells reported in the Sherwood et al. (2017) data set: - 44.8 $^o\!/_{oo}$ for "group 2A and 2B" wells and - 38.5 $^o\!/_{oo}$ for "group 1" wells (Rodriguez and Philp 2010) and -41.1 $^o\!/_{oo}$ (Zumberge et al. 2012). For our analysis, we use the mean of the $\delta^{13}$C ratio (- 46.9 $^o\!/_{oo}$) from three studies where the methane clearly came from horizontal, high-volume fractured shale wells: - 47.0 $^o\!/_{oo}$ for Bakken shale, North Dakota (Schoell et al. 2011), -46.5 $^o\!/_{oo}$ for Barnett shale, Texas (Townsend-Small et

15  al. 2015), and - 47.3 $^o\!/_{oo}$ for Utica shale, Ohio (Botner et al. 2018). Note that several studies have reported mean $\delta^{13}$C ratios for methane from organic-rich shales that are more depleted in $^{13}$C (more negative) than this: -50.7 (Martini et al. 1998) for Antrim shale, Michigan; - 53.3 (McIntosh et al. 2002) and – 51.1 (Schlegel et al. 2011) for New Albany shale, Illinois; and - 49.3 (Osborn and McIntosh 2010) for a Devonian shale in Ohio. However, these shales are not typical of the major shale plays supporting the huge increase in gas production over the past decade.

The average $\delta^{13}$C ratio for methane in the atmosphere in 2005 was - 47.15 $^o\!/_{oo}$ (Schneising et al. 2016), which reflects a flux-weighted mean input of methane with a $\delta^{13}$C ratio of – 53.5 $^o\!/_{oo}$. This flux-weighted mean value is approximately 6.3 $^o\!/_{oo}$ more depleted in $^{13}$C because of fractionation during the oxidation of methane in the atmosphere (Schneising et al. 2016; Sherwood et al. 2017). In our analysis, we use this flux-weighted mean value of – 53.5 $^o\!/_{oo}$. Therefore, the mean value for $D_{A\text{-}FF}$

25  is 9.5 $^o\!/_{oo}$, for $D_{B\text{-}A}$ is 9.0 $^o\!/_{oo}$, and for $D_{A\text{-}SG}$ is 6.6 $^o\!/_{
[revised manuscript text omitted]

[Figure]

[Figure]

[Figure]

Figure 4. A, top: gas blowdown for maintenance on a pipeline in Yates County, NY. While methane is invisible, the cooling caused by the blowdown condenses water vapor, leading to the obvious cloud. Photo courtesy of Jack Ossont.

5  B, bottom: Gas storage tanks receiving natural gas from feeder pipelines before compression for transport in high-pressure pipelines, Haynseville shale formation, Texas. Photo on left was taken with a normal camera. Photo on the right was taken with a FLIR camera tuned to the infra-red spectrum of methane, allowing visualization of methane, which is invisible in the normal camera view and to the naked eye. Photo courtesy of Sharon Wilson.

---

## Author Response (AR1)

July 5, 2019

Prof. Jack Middleburg
Associate Editor
*Biogeosciences*

Dear Jack:

Thank you for your thoughtful comments on my revised submission #bg-2019-131 entitled "Is shale gas a major driver of recent increase in global atmospheric methane." I agree with each of your points and have revised the manuscript further accordingly. The new manuscript follows, with the new changes indicated in edit mode. Specifically, I revised the model presentation to follow each of your points "a" through "e." Note that the new Eq. (7) looks a little different from Eq. (5) in the previous manuscript, but in fact the two equations are equivalent and provide the same results; the difference is simply that some of the terms are reversed in sign, a result of your suggested derivation. I could of course have added a few more algebraic steps to result in an equation identical in appearance to the old Eq. (5), but I saw no useful purpose in doing so.

I very much appreciate your handling of my submission and your very useful feedback.

Best regards,

Robert W. Howarth, Ph.D.
*David R. Atkinson Professor of Ecology*
*and Environmental Biology*

[revised manuscript text omitted]
 after 2007. That is, the inclusion of an estimate for shale gas is matched by changes in the estimated fluxes from biogenic sources and other fossil fuels.~~

Eq. (4) builds on Eq. (3) and reweights the information in Figure 3-A for the difference between most fossil fuels and shale gas, multiplying global mass fluxes for each source by the difference between the $\delta^{13}C$ ratio of each source and the flux-weighted mean for all sources:

$$[ ( B_{W\cancel{N}} - B_{N\cancel{W}} ) * D_{B\text{-}A} ] = + [ ( FF_{W\cancel{N}} - FF_{N\cancel{W}} ) * D_{A\text{-}FF\text{-}A} ) ] - + ( SG * D_{A\text{-}SG\text{-}A} ) = 0$$

(4)

where $D_{B\text{-}A}$, $D_{FF\text{-}A}$, and $D_{SG\text{-}A}$ are the differences in the $\delta^{13}C$ ratio of biogenic emissions, fossil fuels, and shale gas compared to the flux-weighted mean $\delta^{13}C$ ratio for all sources (A). The x-axis of Figure 3-B shows the $\delta^{13}C$ for each source; note that the y-axis is the estimate of the change in emissions for each of these sources that we derive below. Next,  we multiply both sides of Eq. (3) by $D_{B\text{-}A}$ ,

$$[ ( B_W - B_N ) * ( D_{B\text{-}A} ) ] + [ ( FF_W - FF_N ) * ( D_{B\text{-}A} ) ] - SG * ( D_{B\text{-}A} ) = 0 \tag{5}$$

$$\cancel{( B_N - B_W ) * ( D_{B\text{-}A} ) = [ ( FF_N - FF_W ) * ( D_{B\text{-}A} ) ] - ( SG * D_{B\text{-}A} )} \tag{3}$$

Subtracting Eq. (5) from Eq. (4),

$$[ ( FF_W - FF_N ) * ( D_{FF\text{-}A} - D_{B\text{-}A} ) ] - [ SG * ( D_{SG\text{-}A} - D_{B\text{-}A} ) = 0 \tag{6}$$

$$\cancel{0 = [ ( FF_N - FF_W ) * ( D_{A\text{-}FF} + D_{B\text{-}A} ) ] + [ SG * ( D_{A\text{-}SG} + D_{B\text{-}A} )} \tag{4}$$

Rearranging Eq. (6) to solve for SG,

$$SG = - ( FF_{W\cancel{N}} - FF_{N\cancel{W}} ) * ( D_{A\text{-}FF\text{-}A} - + D_{B\text{-}A} ) / ( D_{A\text{-}SG\text{-}A} - + D_{B\text{-}A} )$$

(7)

Note that from Worden et al. (2017), $FF_N$ is 16.4 Tg per year.

Although our expectation is that the methane in shale gas is depleted in [13]C relative to conventional natural gas, the $\delta^{13}C$ ratios for the methane in both conventional gas reservoirs and in shale gas vary substantially, changing with the maturity of the gas and several other factors (Golding et al. 2013; Hao and Zou 2013; Tilley and Muehlenbachs 2013). The large data set of Sherwood et al. (2017) suggests no systematic difference between the average ratio for shale gas and the average for conventional gas. However, some of the data listed as shale gas in that data set are actually for methane that has migrated from shale to reservoirs (Tilley et al. 2011) and therefore may have been partially oxidized and fractionated (Hao and Zou 2013). In other cases, the data appear to come both from conventional vertical wells and shale-gas horizontal wells in the same

region, making interpretation ambiguous (Rodriguez and Philp 2010; Zumberge et al. 2012). Note that in the Barnett shale region, Texas, the $\delta^{13}$C ratio for methane emitted to the atmosphere ( - 46.5 $^\circ/_{oo}$; Townsend-Small et al. 2015) is more depleted than the average for wells reported in the Sherwood et al. (2017) data set: - 44.8 $^\circ/_{oo}$ for "group 2A and 2B" wells and - 38.5 $^\circ/_{oo}$ for "group 1" wells (Rodriguez and Philp 2010) and -41.1 $^\circ/_{oo}$ (Zumberge et al. 2012). For our analysis, we use the mean

5   of the $\delta^{13}$C ratio (- 46.9 $^\circ/_{oo}$) from three studies where the methane clearly came from horizontal, high-volume fractured shale wells: - 47.0 $^\circ/_{oo}$ for Bakken shale, North Dakota (Schoell et al. 2011), -46.5 $^\circ/_{oo}$ for Barnett shale, Texas (Townsend-Small et al. 2015), and - 47.3 $^\circ/_{oo}$ for Utica shale, Ohio (Botner et al. 2018). Note that several studies have reported mean $\delta^{13}$C ratios for methane from organic-rich shales that are more depleted in $^{13}$C (more negative) than this: -50.7 (Martini et al. 1998) for Antrim shale, Michigan; - 53.3 (McIntosh et al. 2002) and – 51.1 (Schlegel et al. 2011) for New Albany shale, Illinois; and

10   - 49.3 (Osborn and McIntosh 2010) for a Devonian shale in Ohio. However, these shales are not typical of the major shale plays supporting the huge increase in gas production over the past decade.

        The average $\delta^{13}$C ratio for methane in the atmosphere (A) in 2005 was - 47.15 $^\circ/_{oo}$ (Schneising et al. 2016), which reflects a flux-weighted mean input of methane with a $\delta^{13}$C ratio of – 53.5 $^\circ/_{oo}$. This flux-weighted mean value is approximately

15   6.3 $^\circ/_{oo}$ more depleted in $^{13}$C because of fractionation during the oxidation of methane in the atmosphere (Schneising et al. 2016; Sherwood et al. 2017). In our analysis, we use this flux-weighted mean value of – 53.5 $^\circ/_{oo}$. Therefore, the mean value for $D_{A\text{-}FF\text{-}A}$ is - 9.5 $^\circ/_{oo}$, for $D_{B\text{-}A}$ is 9.0 $^\circ/_{oo}$, and for $D_{A\text{-}SG\text{-}A}$ is - 6.6 $^\circ/_{
[revised manuscript text omitted]

---

## Author Response (AR2)

Cornell University

Department of Ecology and Evolutionary Biology

July 9, 2019

Prof. Jack Middleburg
Associate Editor
*Biogeosciences*

Dear Jack:

Thank you once again for your careful read of my submission #bg-2019-131. One page 5, line 12, you are correct, and I have changed FFN to FFW. "Schneising et al. (2016)" should have been Schneising et al. (2014); I have corrected this on page 5 line 33 and page 6 line 1. Regarding your comment on page 8, I am not sure I completely agree with you, but I have taken your advice to be more conservative here in any case. I have rewritten the sentence to now read: "Our finding is also consistent with Schneising et al. (2014), who analyzed data from another satellite and concluded that most of the global increase in methane emissions between 2006-2008 and 2009-2011 came from mid latitudes in the northern hemisphere." This statement is well supported by the data they show in their Fig. 2. I have made all three of the changes you suggest for the two tables. And I have no uploaded the new, correct figures. I apologize for having not caught that before.

Please do let me know if you have any further comments or concerns.

Best regards,

Robert W. Howarth, Ph.D.
*David R. Atkinson Professor of Ecology*
*and Environmental Biology*

[revised manuscript text omitted]

---

## Author Response (AR3)

Department of Ecology and Evolutionary Biology

July 11, 2019

Prof. Jack Middleburg
Associate Editor
*Biogeosciences*

Dear Jack:

Thank you the opportunity to make a few further modifications to my submission #bg-2019-131 entitled "Is shale gas a major driver of recent increase in global atmospheric methane." I have corrected the three technical issues you noted, changing the first incorrect reference to Schaefer et al. (2016) and the second incorrect reference to Schwietzke et al. (2016), and I corrected the mis-spelling of Nisbet. Regarding the concern voiced to you by the reviewer on my interpretation of Schneising et al. (2014), I agree that this is not at all important to my paper, and I have taken your advice and removed the two sentences in this paragraph that refer to the Schneising et al. (2014) paper. I do appreciate your willingness to let me as the author have the final say, but you have given me some excellent guidance which I gladly accept.

With best regards,

Robert W. Howarth, Ph.D.
*David R. Atkinson Professor of Ecology*
*and Environmental Biology*

[revised manuscript text omitted]